# McsB forms a gated kinase chamber to mark aberrant bacterial proteins for degradation

Bence Hajdusits[1†], Marcin J Suskiewicz[1†‡], Nikolas Hundt[2†§], Anton Meinhart[1,3], Robert Kurzbauer[1], Julia Leodolter[1], Philipp Kukura[2*], Tim Clausen[1,3*]

[1]Research Institute of Molecular Pathology (IMP), Vienna BioCenter, Vienna, Austria; [2]Physical and Theoretical Chemistry Laboratory, Department of Chemistry, University of Oxford, Oxford, United Kingdom; [3]Medical University of Vienna, Vienna, Austria

**Abstract** In Gram-positive bacteria, the McsB protein arginine kinase is central to protein quality control, labeling aberrant molecules for degradation by the ClpCP protease. Despite its importance for stress response and pathogenicity, it is still elusive how the bacterial degradation labeling is regulated. Here, we delineate the mechanism how McsB targets aberrant proteins during stress conditions. Structural data reveal a self-compartmentalized kinase, in which the active sites are sequestered in a molecular cage. The 'closed' octamer interconverts with other oligomers in a phosphorylation-dependent manner and, unlike these 'open' forms, preferentially labels unfolded proteins. In vivo data show that heat-shock triggers accumulation of higher order oligomers, of which the octameric McsB is essential for surviving stress situations. The interconversion of open and closed oligomers represents a distinct regulatory mechanism of a degradation labeler, allowing the McsB kinase to adapt its potentially dangerous enzyme function to the needs of the bacterial cell.

**\*For correspondence:**
philipp.kukura@chem.ox.ac.uk (PK);
tim.clausen@imp.ac.at (TC)

[†]These authors contributed equally to this work

**Present address:** [‡] Sir William Dunn School of Pathology, University of Oxford, Oxford, United Kingdom; [§] Ludwig-Maximilians-Universität München, Department of Cellular Physiology, Biomedical Center, Planegg, Germany

**Competing interests:** The authors declare that no competing interests exist.

## Introduction

Microorganisms have evolved specialized defense mechanisms for coping with adverse environmental changes. Pathogenic bacteria, for example, employ an intricate stress-response and protein quality control (PQC) machinery to counteract proteotoxic stresses imposed by the immune system and adapt to nutritional limitation, oxidative stress, and elevated temperatures (*Elharar et al., 2014*; *Müller and Weber-Ban, 2019*; *Raju et al., 2012*; *Rowley et al., 2006*). In Gram-positive pathogens, the repressor CtsR, the protein arginine kinase McsB and the protease ClpC:ClpP constitute the core of the stress-response operon (*Derre et al., 1999*; *Kirstein et al., 2007*; *Krüger and Hecker, 1998*). The system comprises both transcriptional and post-transcriptional mechanisms, enabling the rapid expression of heat-shock factors and the elimination of damaged bacterial proteins, respectively. While the former regulatory network of transcription factors has been characterized in detail (*Chaturongakul et al., 2011*; *Schumann, 2003*), the equally important process of removing aberrant proteins is less understood. The protein kinase McsB is critical for both processes, as it coordinates stress signaling and protein degradation pathways.

Compared to eukaryotes, bacteria express a broader range of protein-phosphorylating enzymes that target a wider spectrum of amino acid residues (*Macek et al., 2019*; *Mijakovic et al., 2016*). McsB is a prominent member of the specialized bacterial kinases and has the unique activity of phosphorylating arginine residues (*Fuhrmann et al., 2009*). Major substrates of McsB are the two central heat-shock repressors CtsR and HrcA (*Schumann, 2003*; *Schumann, 2016*). As shown for CtsR, McsB controls the two transcription factors by phosphorylating arginine residues in their DNA-

reading heads, thus promoting their dissociation from DNA and inducing the expression of stress-response genes (*Fuhrmann et al., 2009*; *Schmidt et al., 2014*; *Trentini et al., 2016*). Ultimately, de-repression of CtsR and HrcA leads to an upregulation of potent PQC factors including the McsB kinase itself, the molecular chaperones GroES and GroEL, the disaggregase ClpE and the ClpC:ClpP protease (*Gophna and Ron, 2003*). Aside from this regulatory role, the phosphoarginine (pArg) mark introduced by McsB has a broader function in the cell, serving as a degradation tag that directs client proteins to the ClpC:ClpP protease (*Stannek et al., 2014*; *Trentini et al., 2016*; *Figure 1—figure supplement 1*). Impairment of the pArg-ClpC:ClpP degradation pathway precludes clearance of protein aggregates (*Stannek et al., 2014*) and limits survival under heat-shock conditions (*Trentini et al., 2016*). This pArg-mediated housekeeping function is likely to underlie McsB's pivotal role as a virulence factor, enabling severe human pathogens like *Staphylococcus aureus* and *Listeria monocytogenes* to cope with adverse conditions imposed by the host immune system (*Singh et al., 2015*; *Wozniak et al., 2012*).

On a molecular level, the McsB kinase is composed of a specialized phospho-transferase scaffold for arginine phosphorylation and a dimerization domain aligning two subunits (*Suskiewicz et al., 2019*; *Figure 1a*). In vitro and in vivo data showed that dimer formation is a prerequisite for attaining full kinase activity, most likely by mechanistically coupling the two active sites (*Suskiewicz et al., 2019*). Although the McsB dimer represents the basic functional unit of the protein arginine kinase, various oligomeric states have been reported, ranging from monomers to dimers and even higher order oligomers (*Fuhrmann et al., 2009*; *Jung et al., 2019*; *Kirstein et al., 2005*; *Trentini et al., 2016*). It is not yet clear, however, whether these different oligomeric states play a role in the bacterial heat shock response. Likewise, the molecular features that enable McsB to

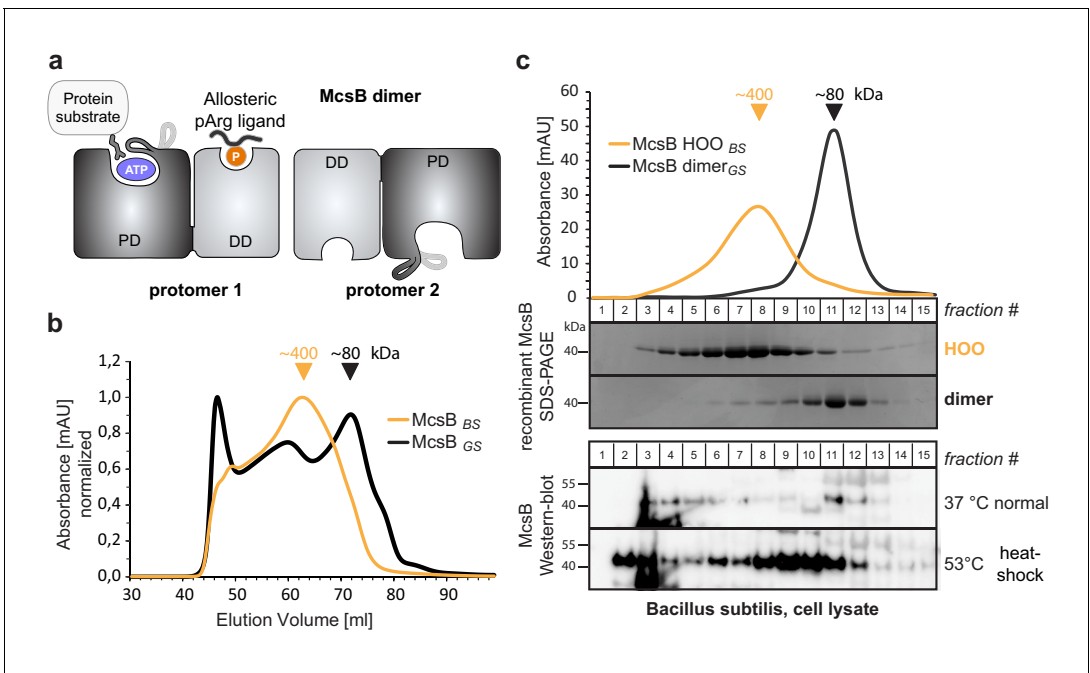

**Figure 1.** McsB$_{BS}$ forms higher order oligomers in vitro and in vivo. (**a**) Schematic picture of the McsB dimer, emphasizing its domain architecture and the location of the catalytic and allosteric sites (PD, phosphotransferase domain; DD, dimerization domain). (**b**) Size exclusion chromatography (SEC) of recombinant McsB$_{BS}$ and McsB$_{GS}$. Triangular markers indicate the protein size at the peaks. (**c**) SEC analysis of lysates of heat-shocked and non-heat shocked *B. subtilis* cell cultures. Comparing the western blots to SDS-PAGE gels of isolated McsB dimer and HOO reveals the different size distributions of McsB under the applied conditions.

The online version of this article includes the following source data and figure supplement(s) for figure 1:

**Source data 1.** Annotated uncropped western blot gels of *Figure 1c*.
**Source data 2.** High quality images of uncropped western blot gels of *Figure 1c*.
**Figure supplement 1.** Degradation pathways in comparison.
**Figure supplement 2.** Higher order oligomers of McsB$_{GS}$.

recognize and bind protein substrates have not been resolved so far. Conceptually, the pArg mark added by McsB is similar to the eukaryotic poly-ubiquitin tag, serving as a degradation signal for general proteolysis. In eukaryotes, the ubiquitin E3 ligases, which carry out the degradation labeling, are tightly controlled enzymes, where substrate selection and/or ligase activity can be adjusted to specific physiological requirements (*Deshaies and Joazeiro, 2009*). Given its important role in deciding about the fate of bacterial proteins, the pArg-tagging activity of McsB should be carefully regulated as well. Proteomics data, however, suggest that the McsB arginine kinase is a rather promiscuous enzyme targeting a large set of substrates, without exhibiting any consensus sequence specificity (*Elsholz et al., 2012*; *Junker et al., 2018*; *Schmidt et al., 2014*; *Trentini et al., 2014*; *Trentini et al., 2016*). It is thus unclear how the McsB kinase selects for aberrant proteins that need to be degraded and distinguishes these damaged molecules from natively folded proteins.

Considering the promiscuity of the McsB kinase and the existence of its different oligomeric states, we studied whether the kinase's oligomerisation status affects its function as a degradation labeler. Our in vitro and in vivo data show that oligomer conversion is critical to adapt the kinase activity of McsB to its PQC function. We found that the pArg-dependent switch in oligomeric state defines McsB's substrate specificity and becomes essential during stress situations to efficiently eliminate damaged proteins. In addition to revealing the central phospho switch that regulates degradation labeling in bacteria, our findings highlight the power of combining structural approaches with novel biophysical methodologies, which we anticipate will reveal similarly complex mechanisms in other biological systems.

## Results

### McsB forms higher order oligomers in vitro and in vivo

Previous structure-function studies have been carried out with the *Geobacillus stearothermophilus* McsB kinase (McsB$_{GS}$), the prevalent model for protein arginine kinases (*Fuhrmann et al., 2013*; *Fuhrmann et al., 2009*; *Jung et al., 2019*; *Suskiewicz et al., 2019*). McsB$_{GS}$ exists mainly as a functional dimer in vitro, but is also capable of forming higher order oligomers, as suggested by native mass spectrometry (*Suskiewicz et al., 2019*) and analytical size exclusion chromatography (SEC) data (*Figure 1—figure supplement 2*). To investigate the biological relevance of the various kinase states, we studied the McsB kinase from *Bacillus subtilis* (McsB$_{BS}$), which exhibits 71% sequence identity to McsB$_{GS}$. After establishing an efficient procedure for recombinant production of McsB$_{BS}$, we performed analytical SEC runs and observed a comparable distribution of oligomeric states. Whereas the profile of McsB$_{GS}$ consists of a prominent dimeric peak at 80 kDa and higher order oligomers (HOOs) with a molecular size of 250–450 kDa, McsB$_{BS}$ mainly consists of large HOOs and exhibits only a minor dimer peak (*Figure 1b*).

To explore whether the higher order kinase forms might be present in vivo, we analyzed *B. subtilis* cells grown under normal and heat-shock conditions. After disrupting the bacteria, we applied the cell lysate to a size exclusion chromatography (SEC) column, separating the bacterial proteins according to size. The western blot profiles of the endogenous protein obtained using an McsB antibody were compared with those of recombinant McsB$_{GS}$ (isolated dimer) and McsB$_{BS}$ (enriched in HOOs), serving as our 80 kDa and 400 kDa size standards, respectively. This analysis indicated that endogenous cellular McsB$_{BS}$ forms assemblies of various sizes (*Figure 1c*). Intriguingly, McsB$_{BS}$ showed different distributions depending on the investigated environmental scenarios. Under standard conditions, the bulk of McsB$_{BS}$ was present as smaller species with sizes comparable to recombinant dimers, whereas heat-shock induced upregulation of markedly larger complexes. The size shift could indicate oligomerization of McsB into species similar to those found for the in vitro reconstituted McsB$_{BS}$ protein. Equally likely, these larger objects could be McsB dimers associated with folded/unfolded substrates, or with substrates whose abundance is up-regulated during heat-shock.

The presence of McsB in fractions at the SEC column's void volume might point to the induced formation of larger oligomers of the protein kinase that may be associated with soluble protein aggregates, its preferred substrates in vivo (*Trentini et al., 2016*). In sum, these data suggest that in the cell, the McsB kinase exists in different complexes, which interconvert in response to changing environmental conditions. We consider it likely that these states include the oligomeric states

recapitulated in vitro, but the nature of McsB associations in vivo should be further probed with other techniques in the future.

## McsB from *B. subtilis* assembles a pArg-stabilized octamer

Despite the polydispersity of the recombinant McsB$_{BS}$ sample, we initiated a structural study of it. For this purpose, we performed secondary SEC runs, enriching the peak fraction of the higher-order oligomers, and subjected this sample to crystallization trials. We obtained crystals diffracting to 2.5 Å resolution, revealing the atomic structure of the McsB$_{BS}$ octamer (*Figure 2a*; *Table 1*). The octameric particle represents a tetramer of dimers, with each dimer resembling the reported McsB$_{GS}$ structure in its subunit contacts, domain arrangement and active site organization (*Figure 2—figure supplement 1A*). The four McsB$_{BS}$ dimers form a molecular cage having the length of one dimer, 110 Å, and a diameter of 80 Å. The phosphorylation chamber has well-defined entrance gates with openings of approximately 25 Å. Potential substrates have to pass the entrance gates in order to reach the active sites, which are buried within the inner chamber (*Figure 2b*, *Video 1*). Notably, none of the residues implicated in substrate binding or catalysis is engaged in inter-dimer contacts, suggesting that the internal active sites are functional. Octamer formation proceeds along the extended edge of the brick-shaped McsB dimer, where two clusters of polar residues enclose a small hydrophobic core. From the polar patches, the inner contact site comprises a dense network of hydrogen bonds woven around arginine residue 194. Strikingly, the electron density revealed that Arg194 is phosphorylated at a terminal amine of its guanidinium group (*Figure 2c*). The pArg194 phospho-guanidinium group is accommodated in a complementarily charged pocket of a neighboring dimer. This pArg receptor site (pR-RS) corresponds to the previously identified pArg binding pocket of McsB$_{GS}$ that enables allosteric stimulation (*Suskiewicz et al., 2019*; *Figure 2—figure supplement 1B*). In the McsB$_{BS}$ octamer, Arg337*, Arg341*, and Ser285* of the molecular neighbor closely interact with the pArg194 phosphoryl group, whereas Asp338* tightly binds to the positively charged nitrogen of the pArg guanidinium function (*Figure 2d*). Given these intimate ion-pair and hydrogen-bonding interactions, we presume that the pArg:pR-RS* tethering, established twice between each pair of aligned dimers, is the major interaction stabilizing the McsB$_{BS}$ octamer. Consistent with this, the pArg:pR-RS* contact accounts for half of the inter-dimer interface, which measures 850 Å$^2$. In conclusion, structural analysis of the protein arginine kinase from *B. subilits* revealed the structural organization of the McsB octamer. It assembles into a self-compartmentalized kinase stabilized by the auto-phosphorylated residue pArg194. Although post-translational modifications have often been implicated in regulating oligomerization, the structure of the pArg194-linked McsB octamer represents, to our knowledge, the first atomic visualization of a defined higher-order oligomer that strictly depends on a specific phosphorylation event.

## McsB exists in a dynamic equilibrium of monomers, dimers and multi-dimers

To better understand the change in oligomer distribution of *B. subtilis* McsB upon heat shock (*Figure 1c*), we reconstituted the McsB$_{BS}$ system in vitro and characterized the switch mechanism using mass photometry (*Young et al., 2018*). We first studied the concentration-dependent behavior of McsB. The improved size resolution of mass photometry as compared to SEC revealed various small molecular weight forms including 1-, 2-, 4-, and 6-mers at protein concentrations from 10 to 500 nM, with the monomer and dimer being predominant at low concentrations (*Figure 3a*). At increasing concentrations, the population shifted toward formation of higher-order oligomers, such as 4-, 6-, and 8-mers. In addition, we observed small amounts of oligomers consisting of 10 and 12 subunits. When we performed mass photometry on the McsB$_{GS}$ orthologue, we observed a less complex protein mixture at lower protein concentrations (*Figure 3b*). The kinase showed a clear preference to dimerize and started to assemble into higher order oligomers only at higher concentrations, of about 50 nM. Please note that the apparent discrepancy in oligomer abundance between SEC and mass photometry is not only due to the different working concentrations. The single-molecule nature of protein quantification with mass photometry also results in different proportions within the size distribution than for the UV absorption signal in SEC. In mass photometry, each oligomer molecule is counted as a single landing event and therefore contributes equally to the height of its respective peak in the mass distribution. In SEC, large oligomers have an absorption signal that is

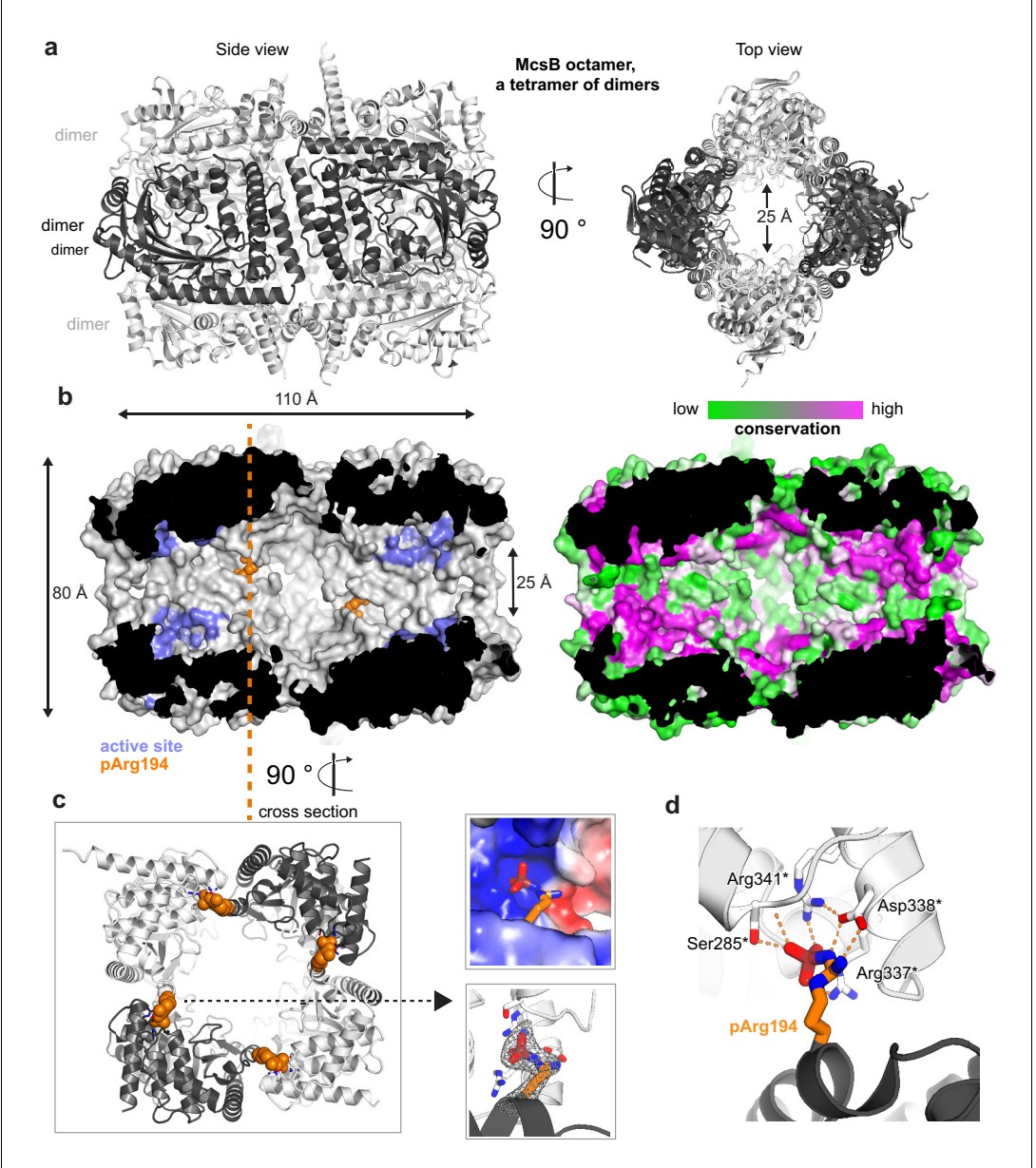

**Figure 2.** Crystal structure of the McsB_BS octamer. (**a**) Ribbon plot of the McsB_BS octamer (side and top views) with alternating dimers colored differently. (**b**) McsB is a self-compartmentalized kinase, as shown in the half-cut surface representation. The location of the active sites within the phosphorylation chamber is highlighted in lilac and the pArg194 clamp in orange. The right panel illustrates the high conservation of the respective active site and interface regions. (**c**) Orthogonal view into the octamer (cross section indicated), with the pArg194 residue highlighted in orange. The right panel illustrates the binding of pArg194 in a complementary charged pocket of the neighboring subunit (top) and its Fo-Fc omit density calculated at 2.5 Å resolution and contoured at 3.0 σ (bottom). (**d**) Structural details of pArg194 locked by intermolecular contacts to the pR-RS of a neighboring protomer (asterisks indicate residues contributed by the adjacent molecule).

The online version of this article includes the following figure supplement(s) for figure 2:

**Figure supplement 1.** Comparison of McsB_BS and McsB_GS.

proportionally stronger the more protein subunits they are composed of, making large species appear more prominent in the SEC profile than smaller ones.

Together, the single-molecule data demonstrate the co-occurrence of various McsB oligomers in a dynamic, concentration-dependent equilibrium (*Figure 3c*). Whereas the octamer, as revealed by the crystal structure, represents a defined and rigid arrangement stabilized by pArg194, the

**Table 1.** Data collection and refinement statistics.

| | Octameric McsB$_{BS}$ (6TV6) |
|---|---|
| Space group | $P2_12_12$ |
| Cell dimensions | |
| $a, b, c$ (Å) | 137.79, 147.64, 81.47 |
| $\alpha, \beta, \gamma$ (°) | 90, 90, 90 |
| | |
| Resolution (Å)[a] | 49–2.5 (2.56–2.5) |
| $R_{meas(I)}$ | 0.146 (1.53) |
| $I/\sigma$ ($I$) | 13.2 (1.2) |
| $CC_{1/2}$ | 0.998 (0.561) |
| Completeness (%) | 99.8 (99.8) |
| Redundancy | 6.7 (6.9) |
| | |
| Resolution (Å) | 49–2.5 |
| No. reflections | 58,139 |
| $R_{work}$ / $R_{free}$ | 0.193 / 0.237 |
| No. atoms | |
| Protein | 11,289 |
| ion | 4 |
| Water | 117 |
| $B$ factors | |
| Protein | 55.6 |
| Ion | 57.7 |
| Water | 51.4 |
| R.m.s. deviations | |
| Bond lengths (Å) | 0.009 |
| Bond angles (°) | 1.083 |

[a]Values in parentheses are for highest-resolution shell.

remaining oligomers are likely to be more heterogeneous in their intermolecular contacts. In this regard, McsB has been shown to modify numerous arginine residues on its own surface (*Elsholz et al., 2012*; *Schmidt et al., 2014*; *Trentini et al., 2014*; *Trentini et al., 2016*). Each of these phospho sites could potentially bind to the pArg-binding pocket of an adjacent McsB dimer to mediate oligomerization. In contrast to the caged 8-mer, these loosely linked, irregular McsB$_{BS}$ assemblies should exist as 'open' kinase forms, in which the active sites are freely accessible. The large number of oligomers identified here is unprecedented among protein kinases and points to a complex enzymatic system, regulating the pArg labeling function. Moreover, the overall distribution of McsB$_{BS}$ and McsB$_{GS}$ oligomers suggest that the formation of multi-dimer assemblies is a common property of this kinase class that might be fine-tuned in evolution to match conditions encountered by various bacterial species.

## McsB oligomers interconvert in a pArg-dependent manner

The crystal structure of McsB$_{BS}$ implies that octamer formation depends on a specific pArg residue, pArg194, which could serve as a phospho-switch controlling protein assembly. To explore the impact of arginine phosphorylation on the stability of the various McsB forms, we incubated the protein samples with YwlE, a pArg-specific phosphatase (*Fuhrmann et al., 2013*). We used McsB$_{BS}$ at a

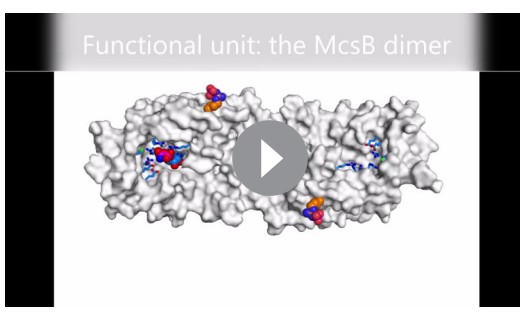

**Video 1.** Illustration of the structural organization of the McsB_BS octamer, which forms a self-compartmentalized protein kinase. The video introduces the basic building block, the McsB dimer, showing the location of the active site and the pArg clamp. After assembling the McsB octamer, the surface presentation emphasizes the self-compartmentalization of the protein kinase, locking the active sites within a phosphorylation chamber. Narrow entry gates restrict access of substrate allowing the selective degradation labeling of unfolded proteins or protein segments.
https://elifesciences.org/articles/63505#video1

concentration of 500 nM, where we could observe 2-, 4-, 6-, 8-, and 10-mers. Upon prolonged incubation with the YwlE phosphatase, we found that higher-order oligomers disappeared while, in parallel, the dimer fraction increased (*Figure 3d*). A notable exception is the McsB octamer, which could not be destabilized by the YwlE phosphatase. These mass photometric data are consistent with the crystal structure of the octamer, where the pArg194 phosphoclamp is deeply buried in the molecular cage and thus inaccessible to YwlE (*Figure 3—figure supplement 1*). To confirm the specific shielding of pArg194, we performed an MS analysis of the YwlE-treated sample, monitoring the phosphorylation status of every McsB arginine residue individually. In accordance with the crystal structure, all pArg residues, except pArg194 and the adjacent pArg190, were quantitatively dephosphorylated (*Figure 3e*). The resistance of McsB octamers to dephosphorylation likely also reflects their stability. Each McsB dimer in the octamer is bound to two other molecules and is thus stabilized compared to irregular multi-dimers, where most units are tethered by single pArg:pR-RS linkages. The high stability of the octamer should limit transient dissociation, which might be a prerequisite for dephosphorylation.

To further explore the stability of the octamer, we analyzed the effect of pArg as a competitive inhibitor of McsB oligomer formation. Given the dynamic interchange of McsB units, we reasoned that free pArg competes with auto-phosphorylated molecules for binding to the pR-RS, thus disturbing the polymerization of McsB dimers. To test this prediction, we incubated auto-phosphorylated McsB with free pArg using concentrations from 0.01 to 5 mM. The addition of pArg selectively destabilized the irregular McsB_BS oligomers, as seen for the tetramer and hexamer (*Figure 3f*). Overall, free pArg shifted the equilibrium toward the isolated monomer and dimer, and to the caged octamer. According to these data, McsB oligomer conversion can be achieved in trans, for example by pArg-labeled products or pArg-carrying effector proteins.

Finally, we tested whether pArg194 stabilized the octamer in a specific manner or, alternatively, induced multimerization of McsB dimers in general. For this purpose, we generated the R194K mutant, which cannot be autophosphorylated at this position. As revealed by mass photometric data, the R194K mutation selectively destabilized the McsB octamer, whereas the relative amounts of open McsB multimers, which can be linked by other pArg sites, were not affected (*Figure 3g*). Moreover, a SEC analysis revealed that mutating the pArg receptor site (R337A/D338A mutant) resulted in destabilization of McsB multimers, such as 4-, 6-, and 8-mers, leading to an accumulation of the dimeric kinase (*Figure 3—figure supplement 2*).

Together, these data indicate that interconversion of McsB oligomers is a pArg-dependent process. Binding of exposed pArg residues to the pR-RS of a molecular neighbor connects McsB dimers to multimers ranging in size from 2- to 12-mers. Among the tethering pArg residues, pArg194 obtains a special role as this phospho residue can promote assembly of the closed octameric cage. In contrast to other higher-order oligomers, the self-compartmentalized octamer persists in the presence of the YwlE phosphatase and pArg-linked effector molecules. We thus presume that in vivo, the interconversion between the different oligomeric states of McsB represents an important function of the PQC kinase.

## McsB dimer and octamer have distinct kinase properties

The shift in equilibrium between open and closed McsB_BS oligomers could represent a functional switch controlling McsB activity. To study the proposed switch in kinase function, we had to reconstitute the octamer and dimer species of the *B. subtilis* McsB_BS and study the two forms separately.

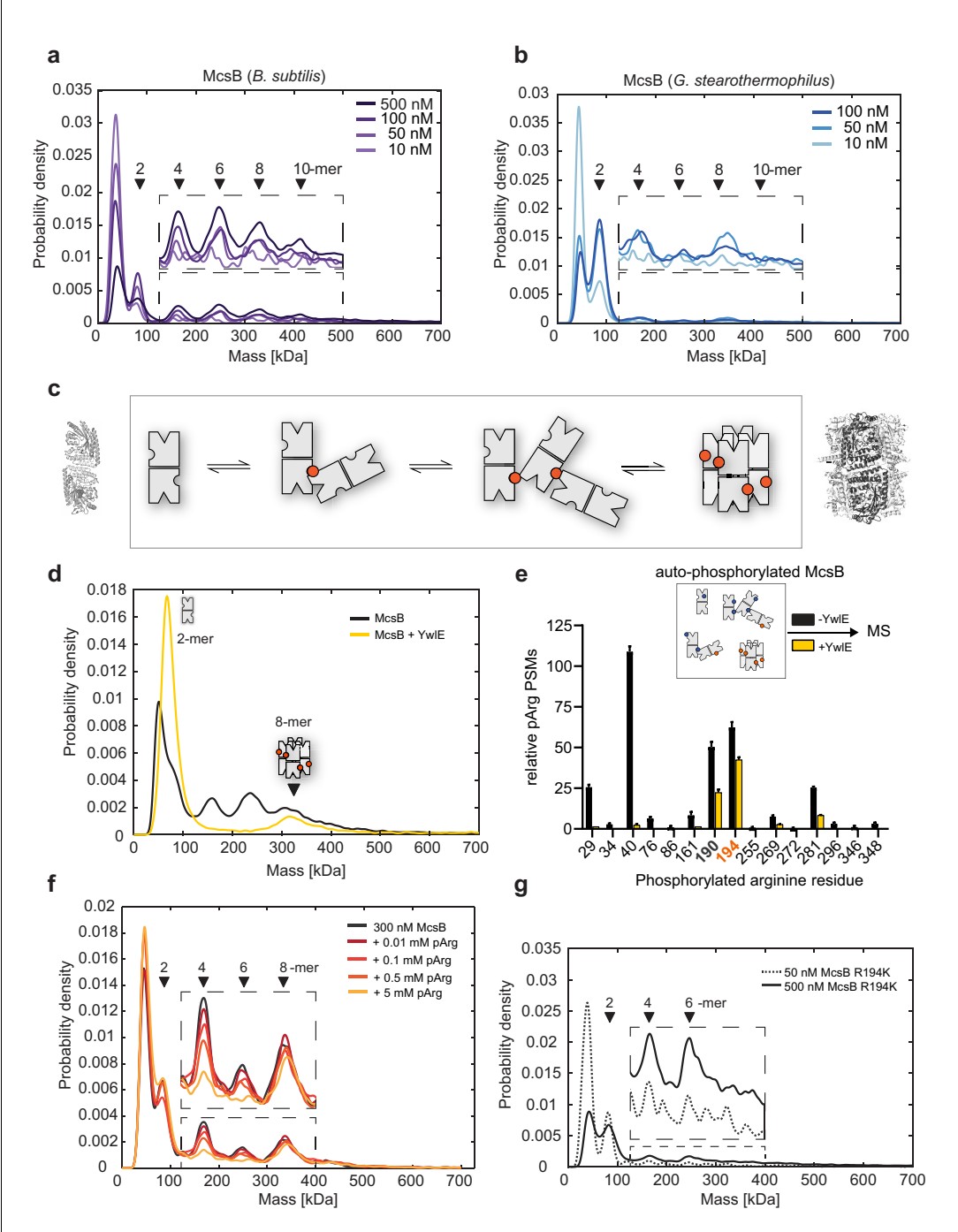

**Figure 3.** McsB_{BS} oligomer conversion depends on a phospho-arginine switch. (a) Distribution of McsB_{BS} oligomers at increasing protein concentrations (10–500 nM). (b) Distribution of McsB_{GS} at increasing protein concentration (10–100 nM). (c) Proposed model of McsB oligomer conversion, with pArg residues represented by colored circles. Structural data are present for dimer and octamer. (d) Mass photometric analysis of the effect of the YwlE phosphatase on McsB_{BS} oligomer conversion. (e) MS data showing that all pArg residues of autophosphorylated McsB are quantitatively removed, except pArg190 and pArg194. The data show the relative number of phosphoarginine PSMs, plotted as mean ± standard deviation from three technical replicates. For absolute number of PSMs as well as all calculations refer to Source Data File. (f) Mass photometric analysis of the effect of free pArg on McsB_{BS} oligomer conversion. (g) Mass photometric analysis of the R194K mutant, showing selective destabilization of the octamer. Size markers are indicated.

The online version of this article includes the following source data and figure supplement(s) for figure 3:

**Source data 1.** Statistics summary of mass photometry experiments.

*Figure 3 continued on next page*

*Figure 3 continued*

**Source data 2.** Mass spectrometry data used for generating *Figure 3e*.
**Source data 3.** Correlation of interferometric contrast and molecular mass.
**Figure supplement 1.** Shielding of pArg190 and Arg194 in the dimer-dimer interface.
**Figure supplement 2.** pArg-binding-deficient mutant R337A/D338A.

Prompted by the mass photometry results, we applied a 'nano-molar' biochemical approach and diluted the McsB$_{BS}$ sample, containing higher-order McsB forms, to a final concentration of 50 nM, thereby favoring the formation of monomers and dimers (*Figure 4a*). The diluted sample was incubated with the YwlE phosphatase. After dephosphorylating all accessible pArg residues, McsB was concentrated, applied to a SEC column, and the dimeric and octameric kinase separated (*Figure 4a*, right panel).

Using the isolated McsB$_{BS}$ forms, we performed radiometric assays monitoring phosphorylation of β-casein, our model substrate. Although the octamer showed residual activity against casein, the

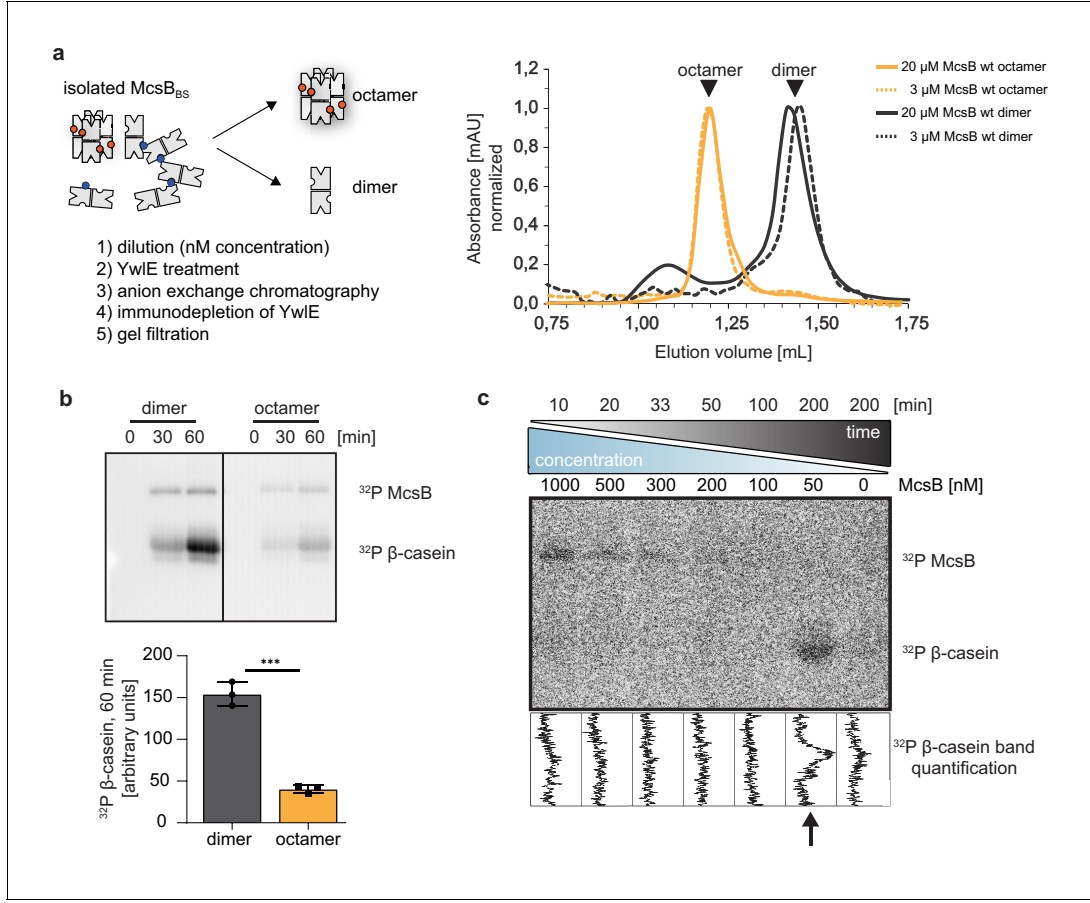

**Figure 4.** McsB$_{BS}$ dimer and octamer exhibit distinct kinase activities. (a) Outline of preparing dimeric and octameric McsB$_{BS}$ (left panel). The established procedure efficiently reconstituted the two *B. subtilis* kinase forms, as seen in the SEC profiles of the separated McsB$_{BS}$ samples (right panel). (b) Radiometric $^{32}$P kinase assays visualizing the activity of dimeric and octameric McsB$_{BS}$ (1 μM) against β-casein (55 μM). ***p≤ 0.001; two-tailed unpaired t test. Data are plotted as mean ± SD (n=3, independent experiments) (c) Radiometric kinase assays using an McsB dilution series. To account for the different enzyme amounts (1000–50 nM), assays were incubated for different times (10–200 min).
The online version of this article includes the following source data for figure 4:

**Source data 1.** Annotated uncropped gels (4b, 4c) and gels used for quantification (4b).
**Source data 2.** High quality images of uncropped gels (4b, 4c) and gels used for quantification (4b).
**Source data 3.** Quantification of gels used for generating the graph in *Figure 4b*.

activity of the dimer was fourfold higher, revealing the preferential targeting of casein by the McsB$_{BS}$ dimer (**Figure 4b**). To further confirm these data, we performed a 'dilution-series' kinase assay, in which McsB$_{BS}$ was sequentially diluted from 1000 to 50 nM (**Figure 4c**). To compensate for the different amounts of the kinase, the reaction times of the enzyme assays were prolonged by the respective factor, ranging from 10 to 200 min. As clearly seen for the casein substrate, McsB gains activity upon dilution. We could only observe substrate phosphorylation when using McsB$_{BS}$ at 50 nM concentration, where oligomers fall apart into dimers (**Figure 3a**). These data suggest that a change in oligomeric state is directly coupled to a switch in the substrate specificity of the protein kinase.

## The octamer of McsB imposes a substrate filter to the kinase function

Because the McsB$_{BS}$ dimer and octamer exhibited distinct activities against casein, we next extended our analysis to additional model substrates. We reasoned that the self-compartmentalized octamer may select for small and unfolded proteins, whereas the dimer may function as a promiscuous kinase. To test this hypothesis, we used a set of model proteins differing in size from 10 to 100 kDa, and representing largely folded, partially folded and unfolded proteins. We first investigated relatively small proteins, in the range of range of 12–25 kDa, showing different degrees of compactness. Radiometric kinase assays revealed that for v-Myc (bHLHZip domain, 12.5 kDa, largely unfolded), the octamer exhibited a higher phosphorylation activity than the dimer. For ComK repressor (Kre) (*B. subtilis* transcription factor of 20 kDa, partially unfolded), the octamer and dimer exhibited comparable kinase activities, and for β-casein (25 kDa, disordered but can oligomerize via interprotein β-sheets), the dimer had a fourfold higher kinase activity as compared to the octamer (**Figure 5a**). Of note, it has previously been shown for another protein cage, the 26S proteasome, that only the N-but not the C-terminus of β-casein can pass through a narrow entry pore (**Berko et al., 2012**). A similar mechanism for the McsB octamer might explain the inefficient modification of β-casein by this oligomer compared to the open dimer. We next studied the phosphorylation of a large 100 kDa model protein, UNC-45 from *Caenorhabditis elegans*. For UNC-45, variants have been developed that differ in size and foldedness and were used to characterize the quality-control E3 ligase UFD-2 (**Hellerschmied et al., 2018**). Compared to the UNC45$_{wt}$ wild-type protein, the UNC45$_{core}$ mutant just comprises the stably folded core region and lacks the intrinsically unstable UCS domain. The dimeric McsB$_{BS}$ was capable of phosphorylating both UNC45 variants (**Figure 5b**). In contrast, the octameric kinase preferentially targeted the full-length construct harboring the disordered UCS domain, whereas the labeling of UNC-45$_{core}$ was very inefficient. In fact, when comparing all tested model substrates, the difference in substrate selectivity was most pronounced for the compact UNC45$_{core}$ protein, which was almost exclusively targeted by the McsB dimer. According to these data, the dimer and octamer of the McsB kinase have complementary substrate specificities. Whereas the dimer phosphorylates proteins in a rather unspecific manner, the octamer preferentially targets unfolded proteins that can pass its narrow entrance gates and enter the kinase chamber.

## Higher order oligomeric McsB represents an auto-activated state

Besides the dimer and the octamer being prominent oligomeric forms of McsB, we next explored the activity of higher-order oligomers (HOOs) comprising 4, 6, 10 and more subunits. We enriched these HOO open forms by selectively destabilizing the caged octamer in the R190A/R194A (RRAA) double mutant (**Figure 5c**). The design of this mutant was based on the observation that pArg190, similarly to pArg194, was resistant to YwlE phosphatase treatment (**Figure 4c**) and could support pArg194 in stabilizing the octameric cage. Assaying the kinase function of the RRAA double mutant revealed that the open HOOs efficiently target the folded model protein UNC45$_{core}$. Importantly, the phosphorylation activity was twofold higher than that of the dimer (**Figure 5d**). Inherent to the dynamic nature of the open oligomer vs dimer, the HOO fraction also contains dimeric McsB (**Figure 5c**). Therefore, considering that a sample enriched in HOO particles exhibits significantly elevated activity, the activity of pure HOOs should be markedly higher than that of the dimer (**Figure 5d**). These data show that the pArg:pR-RS* mediated linkage of McsB subunits exerts a stimulatory effect on kinase activity, which is consistent with the previously described allosteric effect of binding pArg-containing ligands (**Suskiewicz et al., 2019**). Accordingly, McsB auto-activation should be relevant for both open and closed multi-dimers, accelerating the pArg labeling of client proteins.

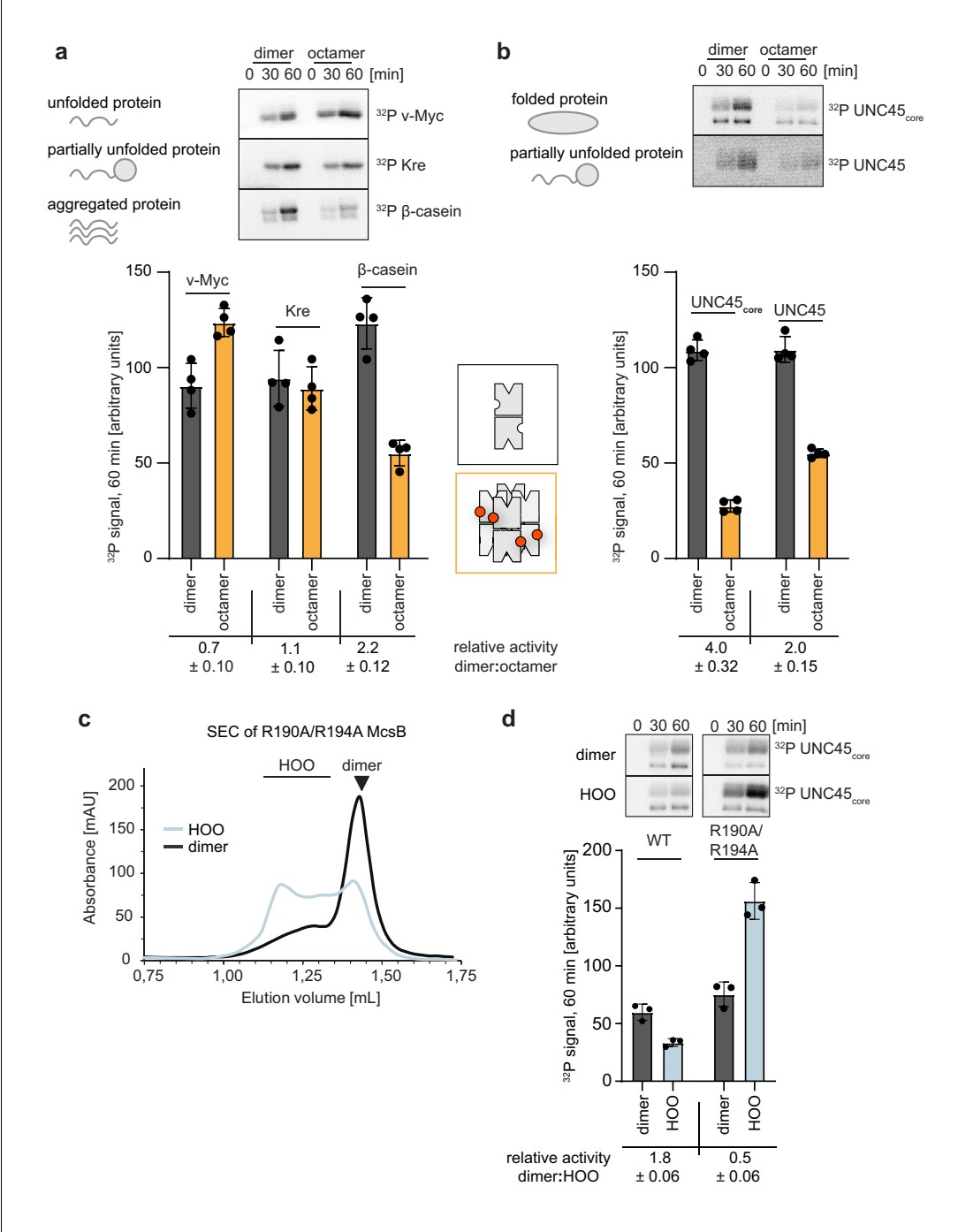

**Figure 5.** A switch in oligomeric state causes a switch in substrate selectivity. (a) Radiometric assays of dimeric and octameric McsB$_{BS}$ (1 μM) using different model substrates. Quantification of activities of dimer (black) and octamer (orange) after 60 min of incubation highlights the distinct substrate preferences (top, representative gel; bottom, quantification of $^{32}$P signal after 60 min, raw data in source data file). Data are plotted as mean ± SD (n=4, independent reactions). Relative activity as mean ± SD. (b) (Top) Kinase assay using the model substrate UNC-45 confirms the substrate filtering role of the octameric cage (top, representative gel; bottom, quantification of $^{32}$P signal after 60 min, raw data in source data file). Data are plotted as mean ± SD (n=4, independent reactions). Relative activity as mean ± SD. (c) Secondary SEC runs of the McsB$_{BS}$ R190A/R194A mutant enriched in either HOOs (higher-order oligomers) or dimers at 20 μM concentration (d) Radiometric kinase assay of wildtype McsB$_{BS}$ and the McsB$_{BS}$ R190A/R194A mutant enriched in either HOOs (higher order oligomers) or dimers (1 μM) against the folded model protein UNC-45$_{core}$ (top, representative gel; bottom, quantification of $^{32}$P signal after 60 min). Data are plotted as mean ± SD (n=3, independent reactions). Relative activity as mean ± SD.

The online version of this article includes the following source data and figure supplement(s) for figure 5:

**Source data 1.** Annotated uncropped gels (5a, b, d) and gels used for quantification (5a, b, d).

*Figure 5 continued on next page*

*Figure 5 continued*

**Source data 2.** High quality images of uncropped gels (5a, b, d) and gels used for quantification (5a, b, d).

**Source data 3.** Quantification of gels used for generating the graph in *Figure 5a,b,d*.

**Figure supplement 1.** Quantitative western blot assay of McsB in *B.subtilis*.

**Figure supplement 1—source data 1.** Annotated uncropped western blot (*Figure 5—figure supplement 1a*).

**Figure supplement 1—source data 2.** High quality images of uncropped western blot (*Figure 5—figure supplement 1a*).

**Figure supplement 1—source data 3.** Calculations used for determining the in vivo McsB levels in *B. subtilis* (*Figure 5—figure supplement 1b*).

The fact that McsB oligomerization is immediately coupled with its activation should be an important mechanistic asset of McsB in the cellular PQC system. Of note, we observed that McsB levels strongly upregulated during stress conditions (*Figure 5—figure supplement 1*), which should favour the formation of higher-order kinase assemblies. Given the ability to undergo multivalent interactions with substrate, we assume that the auto-activated HOO form of McsB is predestined to target protein aggregates.

## McsB octamer formation is crucial for coping with proteotoxic stress conditions

Our in vivo data indicated that McsB oligomers interconvert in a cell-state-specific manner, with higher-order oligomers being enriched under stress conditions (*Figure 1b*). As suggested by in vitro reconstitution experiments, the induced McsB species should mainly represent the closed octamer, the only higher order oligomer that was stable in the presence of YwlE (*Figure 3d*). Moreover, having characterized the octamer as secured kinase that selectively targets misfolded proteins, we wanted to test the role of this particular McsB$_{BS}$ state in the bacterial stress response. To this end, we analyzed the heat-shock-related phenotype of the R194K mutation that specifically destabilized the McsB octamer, leaving the remaining oligomers intact (*Figure 3g*). Strikingly, during heat-shock conditions, the survival of *mcsB(R194K)*-bearing cells was significantly reduced, similar to cells carrying the Δ*mcsB* deletion. After a 2 hr incubation at 53°C, Δ*mcsB* and *mcsB(R194K) B. subtilis* cells suffered severely, as reflected by the strongly reduced number of colony-forming units (CFUs). Whereas 60% wildtype cells survived the imposed heat shock, only 1% to 5% of the Δ*mcsB* and *mcsB(R194K)* mutant cells were thermotolerant (*Figure 6a*). In order to shed more light onto the mechanisms behind this thermosensitive phenotype of *mcsB(R194K) B. subtilis*, we investigated the specific activity of the mutant protein McsB R194K (RK) in a radiometric assay. Compared to the wildtype enzyme, RK showed an increased activity (~1.5 times) against our folded model substrate UNC45$_{core}$ (*Figure 6—figure supplement 1a*). Furthermore, comparing the effect of auto-phosphorylation on the oligomeric distribution between wildtype and RK McsB revealed that dimeric wildtype McsB shifted mostly to the octameric state in the presence of ATP, while this effect was less significant for the RK mutant. Here, the major HOO peak eluted at an apparently smaller molecular size, pointing toward a rapid equilibrium between dimer and open HOO oligomers (*Figure 6—figure supplement 1b*). In sum, these results indicate that the thermosensitive phenotype of the R194K mutant is likely due to a misbalance in the abundance between open HOO forms and the octameric cage, highlighting the functional importance of the pArg194-mediated switch in oligomeric state.

Finally, due to the strong effect that YwlE had on the various McsB oligomers in vitro (*Figure 3d*), we also explored the effect of the pArg phosphatase on McsB function and protein quality control in vivo. For this purpose, we generated a Δ*ywlE B. subtilis* knockout strain and characterized the effect of the phosphatase deletion on thermotolerance. In fact, the YwlE-deficient bacteria showed a strong thermosensitive phenotype. Deletion of the phosphatase almost fully abolished cell survival during heat-shock conditions (*Figure 6—figure supplement 2a*). To test whether the in vivo effect is correlated with the functional state of the McsB kinase, we compared the distribution of McsB sizes in wild-type and Δ*ywlE* cells exposed to heat shock. However, McsB Western blot profiles looked virtually identical (*Figure 6—figure supplement 2b*). Considering the subtle differences in elution profiles of YwlE-resistant closed McsB octamers and YwlE-sensitive open forms (*Figure 6—figure supplement 1b* and *Figure 5c*), as well as the difficulty of distinguishing between McsB oligomers and McsB:substrate associations, this method might not be sensitive enough to capture differences in McsB oligomerization - especially those in terms of topology (closed *vs.* open) rather than weight

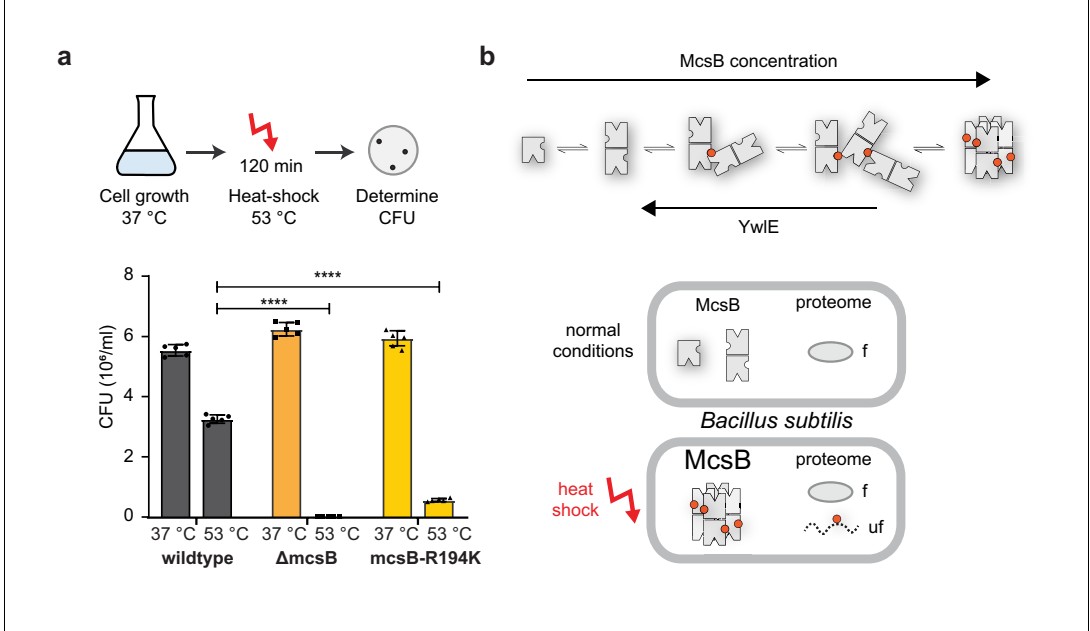

**Figure 6.** Octamer formation of McsB is critical for the heat-shock response. (**a**) In vivo analysis of the R194K McsB oligomerization mutant. Compared to wildtype *B. subtilis* cells, ΔmcsB and R194K mutant cells exhibit a strong thermo-sensitive phenotype. ****p ≤ 0.0001; one-way ANOVA and Tukey's multiple comparison test of heat-shocked samples. Data are plotted as mean ± SD (n=5, biological independent samples). Error bars indicate SD. (**b**) Proposed model illustrating how McsB protein concentration and YwlE activity shape the distribution of McsB oligomers. pArg phospho marks are indicated by red spheres. As shown below, monomer and dimer are prevalent under non-stress conditions. Presumably, their activity on folded proteins (f) can be reversed by the arginine phosphatase YwlE. Under heat-shock, McsB and other components of the stress response machinery are strongly enriched, favoring formation of octameric particles that selectively target misfolded (uf) proteins.

The online version of this article includes the following source data and figure supplement(s) for figure 6:

**Source data 1.** Quantification of colonies used for generating the graph in *Figure 6a*.
**Figure supplement 1.** Comparing wildtype and R194K mutant McsB.
**Figure supplement 1—source data 1.** Annotated uncropped gels (*Figure 6—figure supplement 1a*) and gels used for quantification (*Figure 6—figure supplement 1a*).
**Figure supplement 1—source data 2.** High quality images of uncropped gels (*Figure 6—figure supplement 1a*) and gels used for quantification (*Figure 6—figure supplement 1a*).
**Figure supplement 1—source data 3.** Quantification of gels used for generating the graph in *Figure 6—figure supplement 1a*.
**Figure supplement 2.** Thermosensitive phenotype of Δ*ywlE B.subtilis*.
**Figure supplement 2—source data 1.** Quantification of colonies used for generating the graph in *Figure 6—figure supplement 2a*.
**Figure supplement 2—source data 2.** Annotated uncropped western blot (*Figure 6—figure supplement 2b*).
**Figure supplement 2—source data 3.** High quality images ofu ncropped western blot (*Figure 6—figure supplement 2b*).

- that would be expected based on the clear in vitro observations presented above (*Figure 3d*). The altered equilibrium of McsB forms could thus still account for the harmful effect of *ywlE* deletion. However, it is also possible that the Δ*ywlE* phenotype is explained by the direct role of the phosphatase in rescuing pArg-marked proteins that are still functional and should not be degraded by ClpCP. These two mechanisms are not mutually exclusive, and we suppose that YwlE could limit the collateral damage of McsB to the bacterial cell by affecting both the oligomeric state of the kinase – and thus its substrate specificity – and directly the duration of the phosphorylation signal on substrates. Overall, the in vivo data suggest that the pArg labeling activity of the caged octamer is critical for the PQC function of McsB.

## Discussion

The present study examines the regulation of the bacterial PQC factor McsB, a protein arginine kinase marking proteins for degradation by the ClpCP protease (*Suskiewicz et al., 2019*; *Trentini et al., 2016*). Our data reveal a distinct targeting mechanism for damaged proteins that is

mediated by a controlled, phosphorylation-dependent switch in the oligomeric state of McsB. The complexity we have revealed in the McsB system highlights the power of combining traditional structural approaches with novel biophysical methods. While the crystal structure of the octameric McsB provided the molecular framework for studying the phospho switch that coordinates kinase assembly, mass photometry was crucial for illuminating the heterogeneity of McsB samples and, ultimately, the importance of oligomeric interconversion for function and regulation. McsB is just one example of the many dynamic protein complexes in the cell, most of which exhibit a remarkable versatility in forming functionally distinct assemblies. We thus predict that similar complexity and non-covalent dynamics exist in other biological systems, which are now amenable to the integrative structural and biophysical approaches outlined here.

How do the new findings impact on the function of McsB in the cell? So far, it has been established that McsB constitutes together with ClpC and ClpP the core of the stress response system in Gram-positive bacteria, as also reflected by the thermosensitive phenotype of ΔmcsB cells (*Figure 6a*). In the bacterial PQC system, McsB has to accomplish important tasks. At the onset of the stress response, McsB phosphorylates and inactivates the transcriptional repressors CtsR and HrcA, thereby inducing expression of heat-shock genes, including McsB itself. Our data suggest that the increased levels of McsB lead to the formation of closed octamers (*Figure 6b*), which we show to be the active kinase species responsible for labeling aberrant proteins for degradation. Likewise, upregulation of McsB also favours the formation of higher-order oligomers with enhanced phosphorylation activity but less selective substrate targeting properties (*Figure 5d*). Moreover, our findings indicate that the pArg-dependent degradation system is potentially dangerous to the cell and must be controlled by the phosphatase YwlE. By affecting the oligomeric state (and thus specificity) of McsB and directly dephosphorylating substrates, this enzyme could prevent proteins from being unintendedly directed for ClpCP-mediated degradation. Such 'watchman' or 'proofreader' activity seems to be important, when the activated and less selective HOO McsB kinase is enriched during stress situations. Once the stress stimulus ends, the protein levels of McsB decrease to a level where disassembly of the octamer and other HOOs is favored. Under normal conditions monomer and dimer become the predominant McsB species and while the monomer is enzymatically inactive and may provide a cellular pool of activatable McsB units, the dimer is expected to execute the basal housekeeping activity. Overall, these findings highlight the fine balance in PQC factor activity as required for efficient thermotolerance and to cope with adverse environmental conditions. Our data show that degradation labeling is as carefully regulated in bacteria as it is in eukaryotes. In eukaryotes, ubiquitin E3 ligases are under the control of a variety of measures including the use of adaptor proteins, changes in oligomeric composition and intricate conformational switches (*Lydeard et al., 2013*; *Plechanovová et al., 2011*; *Zheng and Shabek, 2017*). Similarly, the McsB kinase exists in open and closed forms with characteristic substrate preferences, allowing rapid adjustment of its E3-like labeling activity in response to proteotoxic stresses.

Key to controlling McsB is the cage-like organization of the octamer it forms. As visualized by a high-resolution crystal structure, the active sites are sequestered in an internal phosphorylation chamber that is only accessible through lateral entry gates. McsB thus belongs to the family of self-compartmentalized PQC factors, featuring a structural organization similar to that in higher-order proteases, such as the proteasome (*Groll et al., 2000*), ClpP (*Wang et al., 1997*) and DegP (*Krojer et al., 2002*). In these complexes, the buried active sites cannot be accessed by natively folded proteins. Intricate gating and activation mechanisms prevent unwanted action on non-cognate proteins, ensuring the tight control of otherwise dangerous proteolytic machines (*Dong et al., 2019*; *Gatsogiannis et al., 2019*; *Krojer et al., 2008*). Similarly, in the McsB octamer, the small size of the entry gate, which is only 25 Å wide, constitutes a molecular filter favoring the phosphorylation of unstructured polypeptides. Consistent with this, we observed that the caged kinase phosphorylates unfolded proteins in a highly selective and efficient manner. The functional importance of the described McsB phosphorylation chamber is highlighted by the thermosensitive phenotype of the R194K mutation, which impairs octamer stability rather than enzymatic function. To our knowledge, such a strong in vivo phenotype caused by destabilizing a distinct oligomeric state has not been reported for any other PQC factor. We presume that small-molecule compounds blocking octamer formation will abolish the bacterial stress response and cause deleterious effects, as exemplified here for the R194K mutant. Targeting the oligomer conversion of McsB – which is mediated by a

single phosphate group – thus represents an appealing strategy to develop novel antibiotics, interfering with an essential step in the PQC system of bacterial pathogens.

# Materials and methods

**Key resources table**

| Reagent type (species) or resource | Designation | Source or reference | Identifiers | Additional information |
|---|---|---|---|---|
| Gene (*Bacillus subtilis*) | McsB from *B. subtilis* | UniProt | P37570 | - |
| Gene (*Geobacillus stearothermophilus*) | McsB from *G. stearothermophilus* | UniProt | P0DMM5 | - |
| Gene (*Geobacillus stearothermophilus*) | YwlE from *G. stearothermophilus* | UniProt | S0F332 | - |
| Strain, strain background | *Bacillus subtilis*, strain 168, WT | Lab stock | - | - |
| Strain, strain background | *Bacillus subtilis*, strain 168, ΔmcsB | *Suskiewicz et al., 2019* | - | Markerless genomic mutation |
| Strain, strain background | *Bacillus subtilis*, strain 168, R194K | This study | - | Markerless genomic mutation |
| Strain, strain background | *Bacillus subtilis*, strain 168, ΔywlE | Kindly provided by Emmanuelle Charpentier | - | Markerless genomic mutation |
| Recombinant DNA reagent | McsB from *B. subtilis* | *Trentini et al., 2016* | RRID:Addgene_173905 | pET-21a plasmid containing mcsB-His6 from *B. subtilis* |
| Recombinant DNA reagent | McsB from *G. stearothermophilus* | *Suskiewicz et al., 2019* | RRID:Addgene_173906 | pET-21a plasmid containing mcsB-His6 from *G. stearothermophilus* |
| Recombinant DNA reagent | YwlE from *G. stearothermophilus* | *Fuhrmann et al., 2013* | RRID:Addgene_173911 | pET-21a plasmid containing ywlE-His6 from *G. stearothermophilus* |
| Recombinant DNA reagent | McsB from *B. subtilis* R194K | This study | RRID:Addgene_173907 | pET-21a plasmid containing mcsB R194K-His6 from *B. subtilis* |
| Recombinant DNA reagent | McsB from *B. subtilis* R190A/R194A | This study | RRID:Addgene_173908 | pET-21a plasmid containing mcsB R190A/R194A-His6 from *B. subtilis* |
| Recombinant DNA reagent | McsB from *B. subtilis* R337A/D338A | This study | RRID:Addgene_173909 | pET-21a plasmid containing mcsB R337A/D338A-His6 from *B. subtilis* |
| Recombinant DNA reagent | Kre | This study | RRID:Addgene_173912 | pET-21a plasmid containing kre-His6 from *B. subtilis* |
| Peptide, recombinant protein | McsB from *B. subtilis* | This study | NP_387966.1 | C-terminal 6xHis |
| Peptide, recombinant protein | McsB from *G. stearothermophilus* | This study | WP_033017267.1 | C-terminal 6xHis |
| Peptide, recombinant protein | YwlE from *G. stearothermophilus* | This study | WP_033017317.1 | C-terminal 6xHis |
| Peptide, recombinant protein | McsB from *B. subtilis* R194K | This study | NP_387966.1: p.Arg194Lys | C-terminal 6xHis |
| Peptide, recombinant protein | McsB from *B. subtilis* R190A/R194A | This study | NP_387966.1: p.Arg190Ala_ Arg194Ala | C-terminal 6xHis |
| Peptide, recombinant protein | McsB from *B. subtilis* R337A/D338A | This study | NP_387966.1: p.Arg337Ala_ Asp338Ala | C-terminal 6xHis |

*Continued on next page*

*Continued*

| Reagent type (species) or resource | Designation | Source or reference | Identifiers | Additional information |
|---|---|---|---|---|
| Peptide, recombinant protein | Kre | This study | NP_389285.1 | ComK repressor |
| Peptide, recombinant protein | v-Myc | Kindly provided by Konrat lab | NP_045935.1:p. Met1_Gln312del | bHLHZip domain |
| Peptide, recombinant protein | UNC45 from *C. elegans* | Kindly provided by VBCF (Anita Lehner) | NP_497205.1 | C-terminal StrepII |
| Peptide, recombinant protein | UNC45$_{core}$ | Kindly provided by VBCF (Anita Lehner) | NP_497205.1:p. Glu522_Glu961del | C-terminal StrepII |
| Peptide, recombinant protein | beta-casein | SigmaAldrich | C6905 | β-Casein from bovine milk |
| Antibody | Anti-McsB (Rabbit polyclonal) | *Suskiewicz et al., 2019* | - | WB (1:5000) |
| Antibody | Anti-YwlE (Rabbit polyclonal) | *Fuhrmann et al., 2016* | - | Immuno- depletion (0.1 mg/ml) |
| Sequence-based reagent | pET21-mcsB R337A/ D338A-His6_F | This study | - | GCCATTCGAAGAGCGGCTCTCATCAG |
| Sequence-based reagent | pET21-mcsB R337A/ D338A-His6_R | This study | - | CGCTTCGTTCGGTCGCAAAGCGCC |
| Sequence-based reagent | pET21-mcsB R190A/ R194A-His6_F | This study | - | TAAATGCAATTATACCGGCAATT AATCAATTAGGCTTAGTTGTTAG |
| Sequence-based reagent | pET21-mcsB R190A/ R194A-His6_R | This study | - | TTTGCGCAGTTAAAACCAGCG CCGGCAGATGCATCATGACCG |
| Sequence-based reagent | pET21-mcsB R194K-His6_F | This study | - | GATTAATTGCCGGTATAATTTTATTTA TTTGCCTAGTTAAAACCAGCGCCG |
| Sequence-based reagent | pET21-mcsB R194K-His6_R | This study | - | CGGCGCTGGTTTTAACTAGGCAAATAAA TAAAATTATACCGGCAATTAATC |
| Sequence-based reagent | pMAD- mcsB R194K_F | This study | - | ATTATACCGGCAATTAATCA |
| Sequence-based reagent | pMAD- mcsB R194K _R | This study | - | TTTATTTATTTGCCTAGTTAAAACC |
| Other | [gamma-P32] ATP | Hartmann Analytics | #SRP-501 | Volume: 250 uCi |
| Software, algorithm | XDS package | doi:10.1107/ S0907444909047337 | RRID:SCR_015652 | https://xds.mr.mpg.de/ |
| Software, algorithm | PHASER | doi:10.1107/ S0021889807021206 | RRID:SCR_014219 | - |
| Software, algorithm | Phenix | doi:10.1107/ S2059798319011471 | RRID:SCR_014224 | https://www.phenix-online.org/ |
| Software, algorithm | Coot | doi:10.1107/ S0907444904019158 | RRID:SCR_014222 | https://www2.mrc-lmb.cam.ac.uk/ personal/pemsley/coot/ |
| Software, algorithm | Molprobity | doi:10.1002/pro. 3330 | RRID:SCR_014226 | http://molprobity. biochem.duke.edu |
| Software, algorithm | Pymol | other | RRID:SCR_000305 | https://pymol.org/2/ |

## Protein expression and purification

For production of recombinant McsB (*B. subtilis* and *G. stearothermophilus*) and the YwlE phosphatase from *G. stearothermophilus*, we used constructs and procedures described previously (*Fuhrmann et al., 2013*; *Suskiewicz et al., 2019*). In general, protein expression was performed in *Escherichia coli* BL21(DE3) grown in LB-medium supplemented with 50 µg/ml ampicillin. Cultures expressing McsB from *B. subtilis* and mutated variants thereof were grown to an $OD_{600}$ of 1.0 at 37° C, shifted to 18°C and expression was induced upon addition of 0.5 mM IPTG. Cells were harvested after overnight expression and the pellet was resuspended in 50 mM Tris-HCl pH 7.5 and 50 mM NaCl. All proteins were purified by Ni-NTA affinity chromatography using a 5 ml HisTrap column (GE

Healthcare Life Sciences) equilibrated with 50 mM Tris-HCl pH 7.5 and 150 mM NaCl. Bound proteins were eluted with the equilibration buffer supplemented with 250 mM imidazole. Subsequently, the proteins were loaded on a Superdex 200 16/60 column (GE Healthcare Life Sciences) equilibrated with 20 mM Tris-HCl pH 7.5 and 50 mM NaCl. Proteins were concentrated using 30 kDa cut-off Vivaspin columns (Sartorius) and their concentration was determined by their absorbance at 280 nm.

## Mutagenesis

Site-directed mutagenesis of McsB from *B. subtilis* expression constructs were performed using the Q5 Site-Directed Mutagenesis Kit (New England BioLabs) following the manufacturer's procedures. Primers are listed in the Key resources table. The mutation in the *mcsB* locus was introduced into the chromosome of strain *Bacillus subtilis* 168 $mcsB_{R194K}$ as described in *Suskiewicz et al., 2019*, using the mutagenesis primer given in Supplementary Table 2.

## Isolation of dimeric and octameric McsB from *B. subtilis*

The wild-type protein was expressed and initially purified as described above. However, after the Ni-NTA purification the McsB protein (5 mg/ml in 15 ml) was dephosphorylated upon addition of 1 μM of YwlE on ice for 30 min. The sample was loaded on a 6 ml Resource Q column (GE Healthcare Life Sciences) equilibrated with 50 mM HEPES-NaOH pH 7.5 and 50 mM NaCl. Dimeric and octameric McsB were separated in a linear gradient to 500 mM NaCl. Residual YwlE was removed by immuno-depletion using an YwlE antibody (*Fuhrmann et al., 2016*) immobilized on Protein A Dynabeads (Thermo Scientific). The separated oligomeric species were further purified using a Superdex200 10/30 column equilibrated with 50 mM HEPES-NaOH pH 7.5 and 50 mM KCl.

## Analytical size exclusion chromatography

Analytical size exclusion chromatography was performed over a Superdex 200 3.2/300 increase column equilibrated with 50 mM HEPES-NaOH pH 7.5, 50 mM KCl using an Ettan LC system (GE Healthcare Life Sciences). Expected molecular masses were calculated based on the separation of the gel filtration standard (Bio-Rad) by the column.

## Radiometric kinase assay

An aliquot of the McsB kinase (1 μM) was incubated with 12 μM substrate at 37°C in a buffer containing 50 mM HEPES-NaOH pH 7.5, 50 mM KCl, 2% *(v/v)* glycerol, 10 mM β-mercaptoethanol, 7.5 mM $MgCl_2$, 5 mM cold ATP, and 50 μCi $\gamma-32^P$ ATP (Hartmann Analytics). Reactions were quenched at given timepoints upon addition of 5x SDS sample buffer and resolved on a Tris-glycine SDS-PAGE. Gels were dried, exposed to a phosphor imaging plate (BAS-MS 2025, Fuji film) overnight and visualized with a Typhoon Biomolecular imager (GE Healthcare). Quantification was performed by measuring the integrated pixel density of each band using Adobe Photoshop CC.

## Heat-shock assay

A preculture of *B. subtilis* 168 (wildtype or ΔywlE) was grown in LB-medium to stationary phase at 37°C overnight. Five ml were used to inoculate 25 ml LB-medium and the culture was grown to an $OD_{600}$ of 0.5 in duplicates. To evoke a heat-shock, one culture was mixed with 25 ml LB-medium at 53°C and incubated at 53°C for additional 2 hr. The second culture was mixed with 25 ml of LB-medium at 37°C and incubated at room temperature for 2 hr. Cells from both cultures were pelleted, flash frozen in liquid nitrogen, thawed, and resuspended in 500 μL of 50 mM Tris-HCl pH 7.5, 50 mM NaCl supplemented with 250 μg Lysozyme (Sigma), 5 μg DNase I (Roche) and 1x complete Protease Inhibitor Cocktail (Roche). The suspensions were sonicated for 2 min on ice. The lysate was cleared by centrifugation, the total protein concentration was adjusted to 1.8 mg/ml using Bradford reagent (Bio-Rad) and 1 ml aliquots were separated over a Superdex 200 16/60 equilibrated with 20 mM Tris-HCl pH 7.5 and 50 mM NaCl. Fractions of 1.5 ml were collected from which proteins within 1 ml were precipitated with 200 μL cold saturated TCA solution and incubated on ice for around 15 hr. The fractions were washed two times with ice-cold acetone and left drying. Each pellet was resuspended in 30 μL of SDS-PAGE sample buffer and separated on a 4–12% Bis-Tris SDS-PAGE gel (NuPAGE) using 1x MES running buffer (NuPAGE). Immunoblotting was performed using anti-McsB

antibody following standard procedures. Standards used during size exclusion chromatography experiments of cell lysates were composed of 71 µM dimeric McsB *G. stearothermophilus* (dimer) and 71 µM oligomeric McsB from *B. subtilis* purified as described above.

## Cell viability assay

A preculture of each *B. subtilis* 168 strain (WT vs. Δ*mcsB* vs *mcsB*$_{R194K}$) was grown from a single colony in LB-medium to stationary phase at 37°C degrees overnight. 6 ml modified synthetic medium (*Antelmann et al., 1997*) (50 mM Tris-HCl pH 7.5, 15 mM $(NH_4)_2SO_4$, 8 mM $MgSO_4$, 27 mM KCl, 7 mM Na-citrate, 0.6 mM $KH_2PO_4$, 2 mM $CaCl_2$, 1 µM $FeSO_4$, 10 µM $MnSO_4$, 4.5 mM K-glutamate, 0.20% *(v/v)* glucose, 100 µM L-Trypthophan) was inoculated with the preculture to an $OD_{600}$ of 0.05 and grown to 0.4. Each culture was split into halves and one part was transferred to 53°C to evoke heat stress while the other half was kept at room temperature for 2 hr. After the heat-shock, the cultures were diluted 1:1000 using synthetic medium. 50 µL of culture were withdrawn and plated on pre-warmed LB-agar plates and colony forming units (CFU) were determined using ImageJ (version 1.47).

## Quantitative western blot assay

A preculture of *B. subtilis* 168 was grown from a single colony in LB-medium to stationary phase at 37°C degrees overnight. 80 ml of LB-medium was inoculated with the preculture to an $OD_{600}$ of 0.05 and grown to around 0.7. Afterwards, the culture was transferred to 50°C to evoke heat stress for 60 min. Each timepoint (0, 30, and 60 min), 1 ml of culture was collected for the western blot assay and an additional 30 ml was collected for the wet pellet weight determination. The $OD_{600}$ of each culture was determined and the pellets of the western blot samples were resuspended in 80 µL B-Per (Thermo Fisher Scientific) complemented with DNaseI (2 µg/ml), Lysozyme (4 µg/ml) and 1x of complete Protease Inhibitor cocktail (Roche) and lysed for 15 min at room temperature. All samples were $OD_{600}$ corrected to 0.7. SDS-PAGE sample buffer was added in a 1:4 ratio to each sample and 15 µL of each sample was separated on a 12% SDS-PAGE gel (hand-cast/stain-free) using 1x Tris-Glycine running buffer. Immunoblotting was performed using anti-McsB antibody following standard procedures. Intensities were determined by densitometry using ImageJ (version 1.47). The calculations for determining the final McsB concentrations can be found in *Figure 5—figure supplement 1—source data 2*.

## Crystallization and structure determination

Octameric McsB$_{BS}$ was crystallized in a hanging drop, vapor diffusion setup at 100 mg/ml concentration using 200 mM of Mg acetate as reservoir solution. Crystals were grown at 4°C and transferred into a solution containing 120 mM Mg acetate and 40% (v/v) PEG400 used as cryo-protectants. After crystals were flash-frozen in liquid nitrogen, X-ray diffraction data were collected at the beamline X06SA at the Swiss Light Source (Villigen-PSI, Switzerland). Diffraction data were processed and scaled using the XDS package (*Kabsch, 2010*) to a resolution of 2.5 Å. Initial phases were obtained by Molecular Replacement using PHASER (*McCoy et al., 2007*) and the structure of McsB$_{GS}$ (6FH1) as starting model (*Suskiewicz et al., 2019*). The model was improved in iterative cycles of manual building using COOT (*Emsley and Cowtan, 2004*) and refinement with Phenix (*Liebschner et al., 2019*) omitting 5% of randomly selected reflections for calculation of R$_{free}$. Model quality was monitored using MolProbity (*Williams et al., 2018*) and the final model exhibited good stereochemistry with 98% of residues in favored regions of the Ramachandran plot and without any outliers.

## Statistical analysis

Data analysis and statistical calculations of the cell viability assay and radiometric kinase assay were performed in GraphPad Prism 8. Other data analysis was performed in Excel (Microsoft) if not stated otherwise. Excel (version 2013, Microsoft) was used for graph preparations, and they were further modified in Illustrator (version 2020, Adobe).

## Mass spectrometry of McsB

McsB from *B. subtilis* (1 mg/ml) was incubated in triplicates in the presence or absence of 0.2 mg/ml YwlE from *G. stearothermophilus* in 50 mM Tris-HCl pH 7.5, 50 mM NaCl at 37°C for 1 hr in a final

reaction volume of 100 µl. Two mM Vanadate (Sodium Orthovanadate, NEB, pre-incubated for ten minutes at 95°C to dissociate Vanadate oligomers) was added to each reaction to inhibit the YwlE phosphatase. The samples were reduced with 10 mM DTT at 37°C for 60 min and alkylated with 30 mM methyl methanethiosulfonate (MMTS) at room temperature for 45 min. After addition of MMTS, the pH was immediately adjusted to ~seven by addition of 1 M HEPES-NaOH pH 7.2. Samples were digested with 2.5 µg of Trypsin Gold (Mass Spectrometry Grade, Promega, solubilized to 1 µg/µl in 50 mM Acetic Acid) overnight at 37°C. Directly after addition of Trypsin, the pH was adjusted as above.

For LC-MS Analysis, an UltiMate 3000 RSLC nano system was coupled to an Orbitrap FusionLumosTribrid mass spectrometer (Thermo Fisher Scientific). Tryptic peptides were loaded onto a trap column (PepMap C18, 5 mm × 300 µm ID, 5 µm particles, 100 Å pore size, Thermo Fisher Scientific) at a flow rate of 25 µL/min using 0.5% acetic acid (pH 4.5 with $NH_4OH$) as mobile phase to prevent pArg hydrolysis during the removal of salts in the precolumn (*Schmidt et al., 2014*). After 10 min, the trap column was switched in line with the analytical column (PepMap C18, 500 mm × 75 µm ID, 2 µm, 100 Å, Thermo Fisher Scientific). Peptides were eluted by a binary gradient of buffer A and B over a period of 60 min (From 98% A (0.1% (v/v) formic acid) and 2% B (80% (v/v) acetonitrile/0.08% (v/v) formic acid) to 35% B) with a flow rate of 230 nl/min. After electrospray ionization EASY-Spray source (Thermo Fisher Scientific), the MS was operated in data-dependent mode, using a full scan (m/z range 380–2000, nominal resolution of 120,000, target value $4x10^5$) followed by MS/MS scans using an Orbitrap resolution of 30,000 with 3 s cycle time (isolation width 1.2 m/z, mass tolerance 10 ppm). Precursor ions selected for fragmentation (excluding charge state 1, 7, 8, >8) were put on a dynamic exclusion list for 10 s and minimum intensity threshold for precursor selection was set to $2x10^4$. Collision Induced Dissociation with 30% normalized collision energy was used for fragmentation. Additionally, the AGC target was set to $2x10^5$ and the maximum injection time was set to 250 ms.

Raw data were extracted using Proteome Discoverer (version 2.3.0.523, Thermo Scientific) and searched with MS Amanda 2.0 (*Dorfer et al., 2014*) against a combined database of *E. coli* strain BL21(DE3) UniProt Reference Proteome with common contaminants and the respective sequences of *B. subtilis* McsB and *G. stearothermophilus* YwlE added. Methylthio-modification of cysteine was set as a fixed modification. Phosphorylation of serine, threonine, tyrosine, and arginine, oxidation of methionine and deamidation of asparagine and glutamine were set as dynamic modifications. Because tryptic cleavage is impaired at phosphorylated arginine, a maximum of three missed cleavage sites was allowed. The peptide mass deviation was set to five ppm; fragment ion mass deviation to 0.02 Da. MS Amanda score was filtered for >150 and the percolator (*Käll et al., 2007*) was used to filter for PEP < 0.01. All phosphopeptide hits were automatically reanalyzed by phosphoRS (*Taus et al., 2011*) for reliable phosphorylation site analysis (threshold for localization probability > 90%). Quantitative information of McsB peptide spectrum matches (PSMs) was obtained by spectral counting, summing up all spectra associated with a specific match and applying a scaling factor for each sample to each PSM by adjusting the values to normalized spectral counts and subsequently used to determine the relative pArg phosphopeptide spectral counts. The graph depicts the mean of the normalized triplicate measurements (with standard deviation).

## Mass photometry assay

The mass photometry experiments were performed on a custom-built iSCAT microscope similar to the one described in *Cole et al., 2017*; *Young et al., 2018* with a 445 nm laser diode (Lasertack). Coverslips were cleaned by sequential bath sonication in $H_2O$, isopropanol and $H_2O$ for 5 min each. They were then rinsed with $H_2O$, blow-dried in a nitrogen stream and assembled into flow chambers (*Yildiz et al., 2003*). Proteins were diluted into McsB Buffer (20 mM Tris-HCl pH 7.5, 50 mM NaCl) and incubated at room temperature for at least 2 hr in the absence or presence of other compounds (e.g. free phospho-arginine as stated). The flow chambers were loaded with buffer to find a clean area and the optimal focus position, defined as the z-position with maximum root-mean-square deviation of the signal from the glass surface pattern. Subsequently, 20 µL of the protein sample were flushed through the flow chamber and landing was recorded at a frame rate of 1 kHz. The videos were saved with two frames averaged and 3 pixels x 3 pixels binned to their mean to give an effective frame rate of 500 Hz and to reduce the file size. The effective pixel size after binning was 70.2 nm.

## Mass photometry data analysis procedure

The videos of proteins binding to the glass surface were analyzed with the software DiscoverMP (version 2.1.0, Refeyn Ltd). The software detects binding events and determines the respective interferometric scattering contrasts. The user can choose how many frames are averaged for continuous background removal (navg) and can set the thresholds T1 and T2 for the two image filters, which are used to detect the binding events. Filter one is based on T-tests of the pixel intensity fluctuations. As a protein binds to the glass, the pixel intensity changes suddenly. This change is associated with an increase of the filter one score calculated as $-\ln(p)$, where p is the p-value of the T-test comparing pixel values at navg frames before and navg frames after the event. The smallest intensity jump amplitude that exceeds random noise fluctuations and is associated with a binding event is controlled by the value of threshold T1. The signatures of the binding events in interferometric images are radially symmetric. Filter two measures the radial symmetry of all pixel neighborhoods of the interferometric images (*Loy and Zelinsky, 2003*). The lowest symmetry score expected at the center of a peak is defined as threshold T2. Pixel clusters that exceed both thresholds T1 and T2 are used for peak fitting. The amplitude of the peak fit provides an estimate for the interferometric peak contrast. The peak signature (point spread function) is modeled as a superposition of two Sombrero functions multiplied by two Gaussians. The small size of McsB monomers of only 42 kDa made it difficult to detect landing molecules of this species quantitatively above the baseline noise. To determine the optimum filter thresholds (T1 and T2) for the detection of these species and an estimate of the percentage of detected particles, we generated a semi-synthetic movie that used frames from a video recorded with McsB Buffer alone to simulate an experimental background and added simulated point spread functions as landing events that had the expected scattering contrast of a 42 kDa protein (contrast = $2.2 \times 10^{-3}$). The point spread function was modeled as a superposition of two sombrero functions that is convolved with two Gaussian smoothing kernels. This function was the same function used to fit experimental landing events. We then varied navg as well as T1 and T2, ran the analysis procedure, and determined the number of true positive and false positive detections. To determine the maximum number of true positive detections possible at the respective signal-to-noise ratio, we simulated 1000 frames with 100 landing events that were not allowed to overlap closer than 12 pixels spatially and 26 frames temporally. Based on this control simulation, we chose navg = 21, T1 = 1.2 and T2 = 0.15 to process the experimental videos. Using these parameters, the number of true positive detections of monomers in the simulation was $57.2 \pm 4.0\%$ (mean ± standard deviation of 5 simulations) and the number of false positive detections was $12.2 \pm 3.0\%$. For dimers (contrast = $4.3 \times 10^{-3}$), a simulation with the same parameters gave $92.2 \pm 3.0\%$ true positive detections and $3.8 \pm 1.6\%$ false positive detections. The interferometric scattering contrast was converted into molecular mass by calibration with standard proteins (here: protein A – 42 kDa, alcohol dehydrogenase dimer and tetramer – 73.5 and 147 kDa, β-amylase dimer and tetramer – 112 and 224 kDa). Their interferometric scattering contrast was plotted as a function of mass and a line was fitted to the resulting graph (*Figure 3—source data 3*). The slope ($5.023 \times 10^{-5}$ kDa$^{-1}$) and intercept ($1.261 \times 10^{-4}$) of the calibration line were used to convert contrast into mass according to mass = (contrast – intercept) / slope.

## Acknowledgements

We thank all members of the Clausen group, in particular Sabryna Junker, Julian Ehrmann and Renato Arnese, for remarks on the manuscript and discussions. We thank Catherine Lichten for critical reading of the manuscript. We are grateful to Robert Konrat, Borja Mateos and Karin Leodolter for providing purified v-myc(bHLHZip) protein and to the staff at the ESRF (Grenoble, France) and SLS (Villingen, Switzerland), where diffraction data were recorded. We also want to thank the VBCF Mass Spectrometry Facility, and especially Karel Stejskal and Elisabeth Roitinger for technical support. We thank Max Hantke for critical contributions to the mass photometry analysis pipeline and Daniel Cole for the constructions and optimisation of the custom-built mass photometer. The final high-resolution data set was collected at the SLS beamline X06SA. Atomic coordinates and structure factors of the McsB$_{BS}$ crystal structure have been deposited at the Protein Data Bank (PDBe) under accession code 6TV6. This work was supported by a grant from the European Research Council (AdG 694978, to TC and CoG 819593 to PK) and by FFG Headquarter Grant 852936 (to

TC). NH was supported by a DFG return grant HU 2462/3–1. The IMP is supported by Boehringer Ingelheim.

## Additional information

### Funding

| Funder | Grant reference number | Author |
|---|---|---|
| H2020 European Research Council | AdG 694978 | Tim Clausen |
| FFD | 852936 | Tim Clausen |
| H2020 European Research Council | CoG 819593 | Philipp Kukura |
| Deutsche Forschungsgemeinschaft | Return Grant HU 2462/3-1 | Nikolas Hundt |

The funders had no role in study design, data collection and interpretation, or the decision to submit the work for publication.

### Author contributions

Bence Hajdusits, Formal analysis, Validation, Investigation, Visualization, Methodology, Writing - original draft, Writing - review and editing; Marcin J Suskiewicz, Nikolas Hundt, Anton Meinhart, Formal analysis, Validation, Investigation, Visualization, Methodology, Writing - review and editing; Robert Kurzbauer, Julia Leodolter, Investigation; Philipp Kukura, Conceptualization, Supervision, Investigation, Methodology, Writing - review and editing; Tim Clausen, Conceptualization, Supervision, Funding acquisition, Investigation, Visualization, Methodology, Writing - original draft, Project administration, Writing - review and editing

### Author ORCIDs

Marcin J Suskiewicz ![ORCID] https://orcid.org/0000-0002-3279-6571
Nikolas Hundt ![ORCID] https://orcid.org/0000-0001-8217-671X
Tim Clausen ![ORCID] https://orcid.org/0000-0003-1582-6924

### Decision letter and Author response

Decision letter https://doi.org/10.7554/eLife.63505.sa1
Author response https://doi.org/10.7554/eLife.63505.sa2

## Additional files

### Supplementary files

• Transparent reporting form

### Data availability

Structure factor amplitudes and coordinate files have been deposited in the Protein Data Bank under the accession number 6TV6.

The following dataset was generated:

| Author(s) | Year | Dataset title | Dataset URL | Database and Identifier |
|---|---|---|---|---|
| Suskiewicz MJ, Hajdusits B, Meinhart A, Clausen T | 2021 | Octameric McsB from Bacillus subtilis. | https://www.rcsb.org/structure/6TV6 | RCSB Protein Data Bank, 6TV6 |

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
