## [Decision Letter]

**Acceptance summary:**

The authors describe the regulation of an arginine kinase responsible for regulating bacterial stress response. They demonstrate that while the minimal unit for activity is a dimeric form that couples catalysis and substrate binding, specific substrate recognition is mediated by an octamer (tetramer of dimers) that forms a chamber in which the active site is accessible to only certain substrates – those with unstructured regions, expected for a quality control response. Octamer formation requires phosphorylation of a specific arginine in the kinase. This is a fascinating mechanism of substrate control that should be of broad interest to readers interested in quality control, specificity and allostery.

**Decision letter after peer review:**

Thank you for submitting your article "McsB forms a gated kinase chamber to mark aberrant bacterial proteins for degradation" for consideration by *eLife*. Your article has been reviewed by 2 peer reviewers, one of whom is a member of our Board of Reviewing Editors, and the evaluation has been overseen by John Kuriyan as the Senior Editor. The reviewers have opted to remain anonymous.

The reviewers have discussed the reviews with one another and the Reviewing Editor has drafted this decision to help you prepare a revised submission.

Summary:

The experiments are well performed and described. The data, including the in vitro reconstitution experiments with the YwlE phosphatase and the mass spec analysis of the R194K mutant, are convincing that pArg194 is essential for octamer formation, and that the octamer confers specificity for unstructured substrates.

Essential revisions:

1. The R194K data argue that the octamer is important for the stress response. However, given the SEC and mass photometry data, it isn't clear that these experiments demonstrate that only the octamer is stable vs. other species higher than dimer. Can they be sure that the non-octamer oligomers are not important in some other aspect of MscB biology? One can imagine that even a poorly defined but stable oligomer might provide some enhancement of catalytic efficiency simply be enhancing local concentration, for example. Also, are there any estimates available for the cellular concentrations of McsB under normal and stress conditions?

2. Line 374: it is noted that YwlE counteracts McsB autophosphorylation, and the discussion implies that it also regulates octamer formation. Is there evidence for this? What is the phenotype of YwlE knockout? Do they see a different distribution of McsB oligomers in YwlE knockout cells?

3. Figure 1c is not well organized, mainly in that it is not clear which gel/blot at the bottom are the absorbance curves related to. If the absorbance curves in Figure 1c are basically the same as the ones in Figure 1b, it would be simpler to just combine 1b and 1c. "HOO" is not defined. Also, the size of the high-order oligomer is smaller than the marker by at least 2 fractions, and there also exists McsB in fractions 2-5, indicating some oligomers of much larger sizes. Are these aggregates?

4. The conclusion that "the overall distribution of McsBBS and McsBGS oligomers suggest that the formation of multi-dimer assemblies is a common property of this kinase class" is not well supported by data since McsGS was shown to form mostly dimers.

5. The R194K mutation clearly alters the heat shock response (Figure 6), but is it possible that it diminishes the enzymatic activity of the dimer form as well as oligomerization? The phosphorylation activity of the mutant should be tested.

6. McsB auto-phosphorylates and Figure 4b shows the auto-phosphorylation band after 30-60 min of reaction. Is it possible that dimers are phosphorylated and converted to octamers after 30 min? Can the authors clarify this by checking the oligomer state after reaction?

[Editors' note: further revisions were suggested prior to acceptance, as described below.]

Thank you for resubmitting your work entitled "McsB forms a gated kinase chamber to mark aberrant bacterial proteins for degradation" for further consideration by *eLife*. Your revised article has been evaluated by Volker Dötsch (Senior Editor) and a Reviewing Editor.

The manuscript has been improved but there are some remaining issues that need to be addressed, as outlined below:

The authors have performed new experiments and responded carefully to the original reviews. Overall the data support the idea that the octamer helps to specify activity toward unstructured substrates, the main take home of the paper. There are, however, a few issues that need to be clarified.

1. Figure 1: HOO (Higher-order oligomer) needs to be defined early in the text and also in the caption to Figure 1.

2. Figure 1c: The two curves are size standards from the different bacterial species, as explained in the text, but this should also be in the caption, which is otherwise very confusing. Critically, the western blot is claimed to show different oligomers of McsB(BS), but this doesn't distinguish between oligomers vs. simply having differently sized substrates bound to it; they clearly don't have the distribution of the recombinant protein shown in gold. In fact, the modest shift in SEC vs. the dimer peak could suggest the latter. In any case, it does not seem to correspond to octamers; these, as these authors note, could be shifted way up when bound to a substrate, but the western blot evidence is not strong. It might be helpful to perform negative stain EM on these fractions.

3. New data on RRAA mutant: In Figure 5c, d it isn't clear what they are using for "HOO" – is it a particular fraction or the entire HOO profile. That is, does this sample still contain dimer? This should be clarified.

4. The δ ywlE heat shock experiments on cell viability are convincing. The size distribution could not be assessed per Figure 6Sb, but why weren't these done under heat shock conditions – wouldn't this make the difference more obvious? Also, what are the higher bands ~55kDa in the Figure 6 supplement 2 western blots?

---

## [Author Response]

Revisions for this paper:1. The R194K data argue that the octamer is important for the stress response. However, given the SEC and mass photometry data, it isn't clear that these experiments demonstrate that only the octamer is stable vs. other species higher than dimer. Can they be sure that the non-octamer oligomers are not important in some other aspect of MscB biology? One can imagine that even a poorly defined but stable oligomer might provide some enhancement of catalytic efficiency simply be enhancing local concentration, for example. Also, are there any estimates available for the cellular concentrations of McsB under normal and stress conditions?

The referees raise an important point regarding the functional relevance of non-octameric oligomers. To this end, we generated the R190A/R194A (RRAA) double mutant, allowing us to reduce the residual amount of octameric form still present in the R194K (RK) mutant. We then performed a comparative analysis of wildtype and RRAA McsB, characterizing the activity of respective dimer and higher-order oligomers (HOOs) against the folded model substrate, UNC-45core. These new data (Figure 5c, d) demonstrate that multimerization enhances the catalytic efficiency of McsB, as now detailed in the following paragraph:

“Besides the dimer and the octamer being prominent oligomeric forms of McsB, we next explored the activity of higher-order oligomers comprising 4, 6, 10 and more subunits. We enriched these HOO open forms by selectively destabilizing the caged octamer in the R190A/R194A (RRAA) double mutant (Figure 5c). The design of this mutant was based on the observation that pArg190, similarly to pArg194, was resistant to YwlE phosphatase treatment (Figure 4c) and could support pArg194 in stabilizing the octameric cage. Assaying the kinase function of the RRAA double mutant revealed that the open HOOs efficiently target the folded model protein UNC45core. Importantly, the phosphorylation activity was 2-fold higher than that of the dimer (Figure 5d). These data show that the pArg:pR-RS* mediated linkage of McsB subunits exerts a stimulatory effect on kinase activity, which is consistent with the previously described allosteric effect of binding pArg-containing ligands (Suskiewicz et al., 2019). Accordingly, McsB auto-activation should be relevant for both open and closed multi-dimers, accelerating the pArg labelling of client proteins.” (Page 12, Results)

As suggested, we also estimated the amounts of McsB in the bacterial cell. Although it is difficult to derive exact absolute numbers, a quantitative Western Blot analysis of normal and heat-shocked *B. subtilis* cells indicated that the protein concentration of McsB increases almost 10-fold upon heat-shock. These data, which are shown in the new Figure 5 —figure supplement 1 further confirm the proposed model of regulation and are described in the Results paragraph below, as well as in the Discussion of McsB regulation (point 2 and 3).

“The fact that McsB oligomerization is immediately coupled with its activation should be an important mechanistic asset of McsB in the cellular PQC system. Of note, we observed that McsB levels got strongly upregulated during stress conditions (Figure 5 —figure supplement 1), which should favour the formation of higher-order kinase assemblies. Given the ability to undergo multivalent interactions with substrate, we assume that the auto-activated HOO form of McsB is predestined to target protein aggregates.” (Page 13, Results)

2. Line 374: it is noted that YwlE counteracts McsB autophosphorylation, and the discussion implies that it also regulates octamer formation. Is there evidence for this? What is the phenotype of YwlE knockout? Do they see a different distribution of McsB oligomers in YwlE knockout cells?

To evaluate the relevance of YwlE for protein quality control in *B. subtilis*, we characterized the thermotolerance of ∆ywlE cells. We observed that deletion of the pArg phosphatase had a toxic effect when cells were exposed to heat-shock, however, did not impact growth under normal conditions. We did not observe an effect on the distribution of McsB oligomers, which could point towards YwlE being directly involved in rescuing falsely labeled substrates from degradation by ClpCP protease. On the other hand, our available methods might not be suitable to detect a difference in oligomeric distribution. For example, with our method it is impossible to distinguish between McsB oligomers and McsB associated with its substrates such as soluble aggregates. Therefore, we would not fully exclude the possibility – strongly suggested by our in vitro results (Figure 3d, e) – that YwlE might have an impact on the oligomeric distribution of McsB. The new results are shown in Figure 6 —figure supplement 2 and discussed in detail in the following paragraphs:

“Finally, due to the strong effect that YwlE had on the various McsB oligomers in vitro (Figure 3d), we also explored the effect of the pArg phosphatase on McsB function and protein quality control in vivo. For this purpose, we generated a ΔywlE *B. subtilis* knockout strain and characterized the effect of the phosphatase deletion on thermotolerance. In fact, the YwlE-deficient bacteria showed a strong thermosensitive phenotype. Deletion of the phosphatase almost fully abolished cell survival during heat-shock conditions (Figure 6 —figure supplement 2a). To test whether the in vivo effect is correlated with the functional state of the McsB kinase, we compared the distribution of McsB sizes in wild-type and ΔywlE cells exposed to heat shock. However, McsB Western blot profiles looked virtually identical (Figure 6 —figure supplement 2b). Considering the subtle differences in elution profiles of YwlE-resistant closed McsB octamers and YwlE-sensitive open forms (Figure 6 —figure supplement 1b and Figure 5c), as well as the difficulty of distinguishing between McsB oligomers and McsB:substrate associations, this method might not be sensitive enough to capture differences in McsB oligomerization – especially those in terms of topology (closed vs. open) rather than weight – that would be expected based on the clear in vitro observations presented above (Figure 3d). The altered equilibrium of McsB forms could thus still account for the harmful effect of ywlE deletion. However, it is also possible that the ΔywlE phenotype is explained by the direct role of the phosphatase in rescuing pArg-marked proteins that are still functional and should not be degraded by ClpCP. These two mechanisms are not mutually exclusive, and we suppose that YwlE could limit the collateral damage of McsB to the bacterial cell by affecting both the oligomeric state of the kinase – and thus its substrate specificity – and directly the duration of the phosphorylation signal on substrates.” (Page 14; Results)

“Our data suggest that the increased levels of McsB lead to the formation of closed octamers (Figure 6b), which we show to be the active kinase species responsible for labelling aberrant proteins for degradation. Likewise, upregulation of McsB also favours the formation of higher-order oligomers with enhanced phosphorylation activity but less selective substrate targeting properties (Figure 5d). Moreover, our findings indicate that the pArg-dependent degradation system is potentially dangerous to the cell and must be controlled by the phosphatase YwlE. By affecting the oligomeric state (and thus specificity) of McsB and directly dephosphorylating substrates, this enzyme could prevent proteins from being unintendedly directed for ClpCP degradation. Such “watchman” or “proofreader” activity seems to be important, when the activated and less selective HOO McsB kinase is enriched during stress situations. Once the stress stimulus ends, the protein levels of McsB decrease to a level where disassembly of the octamer and other HOOs is favored. Under normal conditions monomer and dimer become the predominant McsB species and while the monomer is enzymatically inactive and may provide a cellular pool of activatable McsB units, the dimer is expected to execute the basal housekeeping activity.” (Page 16, Discussion)

3. Figure 1c is not well organized, mainly in that it is not clear which gel/blot at the bottom are the absorbance curves related to. If the absorbance curves in Figure 1c are basically the same as the ones in Figure 1b, it would be simpler to just combine 1b and 1c. "HOO" is not defined. Also, the size of the high-order oligomer is smaller than the marker by at least 2 fractions, and there also exists McsB in fractions 2-5, indicating some oligomers of much larger sizes. Are these aggregates?

Following the advice of the referees, we reorganized Figure 1. We changed the color coding in panel (c) to better illustrate which Western Blot belongs to which SEC profile. Moreover, we adapted panel (b) to show the comparable size distribution of McsBGS and McsBBS, as observed in the primary SEC runs. The SEC profiles of isolated HOO and dimeric McsBBS are illustrated now in (c), whereas the corresponding secondary SEC runs of McsBGS are shown in Figure 1 —figure supplement 2. With regards to SEC fractions 2-5, we presume that these represent McsB multimers bound to misfolded and soluble aggregated proteins and therefore we adapted the corresponding Results section of Figure 1c. It now reads as follows:

“Under standard conditions, the bulk of McsBBS was present as smaller species, presumably monomers and dimers, whereas heat-shock induced an upregulation of markedly larger particles, at least some of which could correspond to oligomers equivalent to those formed by the in vitro reconstituted McsBBS protein. The presence of McsB in fractions corresponding to an even larger size might point to its association with soluble protein aggregates as substrates. In sum, these data suggest that in the cell, the McsB kinase exists in different states, which interconvert in response to changing environmental conditions. This variation likely includes changes in the oligomeric state recapitulated in vitro.” (Page 6, Results)

4. The conclusion that "the overall distribution of McsBBS and McsBGS oligomers suggest that the formation of multi-dimer assemblies is a common property of this kinase class" is not well supported by data since McsGS was shown to form mostly dimers.

We have to apologize that the oligomerization behavior of McsBGS has not been properly described. As discussed under Point 3, we have added respective data to Figure 1b, illustrating the comparable SEC profiles (oligomer distribution) of the two McsB kinases. We also adapted the corresponding paragraph:

“After establishing an efficient procedure for recombinant production of McsBBS, we performed analytical SEC runs and observed a comparable distribution of oligomeric states. Whereas the profile of McsBGS consists of larger oligomers with a molecular size of 250-450 kDa and a prominent dimeric peak at 80 kDa, McsBBS mostly lacks the dimeric fraction. (Figure 1b).” (Page 6, Results)

5. The R194K mutation clearly alters the heat shock response (Figure 6), but is it possible that it diminishes the enzymatic activity of the dimer form as well as oligomerization? The phosphorylation activity of the mutant should be tested.

The referees raise another important point that relates to the kinase activity of the R194K (RK) mutant, hindered in forming the octameric cage. Given the intricate allosteric signaling within the McsB dimer, it is indeed conceivable that any mutation alters kinase activity in cis and that the thermosensitive phenotype is due to a change in phosphorylation efficiency rather than by altered oligomerization properties. To exclude such allosteric side effect, we compared the enzymatic properties of the RK mutant and wildtype enzyme. These data show that the RK mutant exhibit full arginine phosphorylation activity (Figure 6 —figure supplement 1a). However, in contrast to the wildtype protein, auto-phosphorylation does not induce formation of the closed octamer and thus the net kinase activity of the RK mutant against the folded model protein (UNC-45core) is markedly higher (Figure 6 —figure supplement 1b). These new data, together with the thermosensitive phenotype of the ∆ywlE strain, suggest that it is the loss of protective effect of the kinase, disturbing the protein quality control system. We presume that formation of the closed octamer is important to prevent arginine phosphorylation of folded proteins and thus reduce the number of falsely labelled clients. The adapted part in the Results section reads as follows:

“In order to shed more light onto the mechanisms behind this thermosensitive phenotype of mcsB(R194K) *B. subtilis*, we investigated the specific activity of the mutant protein McsB R194K (RK) in a radiometric assay. Compared to the wildtype enzyme, RK showed an increased activity (~1.5 times) against our folded model substrate UNC45core (Figure 6 —figure supplement 2a). Furthermore, comparing the effect of auto-phosphorylation on the oligomeric distribution between wildtype and RK McsB revealed that dimeric wildtype McsB shifted mostly to the octameric state in the presence of ATP while this effect was less significant for the RK mutant. Here, the major HOO peak eluted at an apparently smaller molecular size, pointing towards a rapid equilibrium between dimer and open HOO oligomers (Figure 6 —figure supplement 2b). In sum, these results indicate that the thermosensitive phenotype of the R194K mutant is likely due to a destabilization of the closed McsB octamer potentially leading to a misbalance in the abundance between open HOO forms and the octameric cage, highlighting the functional importance of the pArg194-mediated switch in oligomeric state.” (Page 13, Results)

6. McsB auto-phosphorylates and Figure 4b shows the auto-phosphorylation band after 30-60 min of reaction. Is it possible that dimers are phosphorylated and converted to octamers after 30 min? Can the authors clarify this by checking the oligomer state after reaction?

This is another excellent point raised by the referees. In fact, the referees are right. As discussed under Point 5, during the kinase assay, the McsB dimer becomes autophosphorylated and starts to form various oligomers. Whereas the closed octamer auto-inhibits the activity of wildtype McsB against folded proteins, formation of the caged octamer is mostly prevented by the RK mutation, thus resulting in higher net phosphorylation activity (Figure 6 —figure supplement 1a). Oligomers formed during the reaction were visualized by SEC (Figure 6 —figure supplement 1b), showing an overall shift to higher order multimers. Notably, the relative amount of induced octamer is markedly higher in wt McsB than that of the HOO in the RK mutant. These data are consistent with our mass photometry results indicating that the caged octamer is a relatively stable complex, whereas the open multi-dimers are transiently formed, enabling their rapid recycling into dimer and monomers when protein levels are reduced. Taken together, our data suggest that it is the dysregulated kinase function of the RK mutant that causes the cytotoxic effect during stress conditions. Presumably, the elevated activity against folded proteins cannot be sufficiently counteracted by the YwlE phosphatase leading to an uncontrolled PQC system. In fact, a similar thermosensitive effect was observed in the ΔywlE deletion strain that also results in imbalanced McsB activity. To account for these findings, we have added a new paragraph in the Results section, and it can be found under point 5.

[Editors' note: further revisions were suggested prior to acceptance, as described below.]

The authors have performed new experiments and responded carefully to the original reviews. Overall the data support the idea that the octamer helps to specify activity toward unstructured substrates, the main take home of the paper. There are, however, a few issues that need to be clarified.1. Figure 1: HOO (Higher-order oligomer) needs to be defined early in the text and also in the caption to Figure 1.

We apologize that we did not include a definition of HOO in the manuscript, neither in the text nor figure legend.

The text in the Results section was changed accordingly and now reads as follows:

“Whereas the profile of McsBGS consists of a prominent dimeric peak at 80 kDa and higher-order oligomers (HOOs) with a molecular size of 250-450 kDa, McsBBS mainly consists of large HOOs and exhibits only a minor dimer peak (Figure 1b).

To explore whether the higher-order kinase forms might be present in vivo, we analyzed *B. subtilis* cells grown under normal and heat-shock conditions. After disrupting the bacteria, we applied the cell lysate to a size exclusion chromatography (SEC) column, separating the bacterial proteins according to size. The Western blot profiles of the endogenous protein obtained using an McsB antibody were compared with those of recombinant McsBGS (isolated dimer) and McsBBS (enriched in HOOs), serving as our 80 kDa and 400 kDa size standards, respectively.” (Results, page 6)

For clarification, we additionally adapted the figure legend of Figure 1:

“(b) Size exclusion chromatography (SEC) of recombinant McsBBS and McsBGS. Triangular markers indicate the size of the peaks. (c) SEC analysis of heat-shocked and non-heat shocked *B. subtilis* cell lysates. Comparing the Western blots to SDS-PAGE gels of isolated McsB dimer and HOO reveals the different size distributions of McsB under the applied conditions.”

2. Figure 1c: The two curves are size standards from the different bacterial species, as explained in the text, but this should also be in the caption, which is otherwise very confusing. Critically, the western blot is claimed to show different oligomers of McsB(BS), but this doesn't distinguish between oligomers vs. simply having differently sized substrates bound to it; they clearly don't have the distribution of the recombinant protein shown in gold. In fact, the modest shift in SEC vs. the dimer peak could suggest the latter. In any case, it does not seem to correspond to octamers; these, as these authors note, could be shifted way up when bound to a substrate, but the western blot evidence is not strong. It might be helpful to perform negative stain EM on these fractions.

We thank the Editors and Referees for raising this point. We acknowledge that our SEC and Western blot-based analysis of endogenous McsB present in the bacterial cell lysate does not conclusively prove the presence of any specific higher oligomer in the cell because it lacks the required resolution to distinguish between homo-oligomers and other associations, as aptly pointed out.

Moreover, considering the difference between endogenous McsB in the complex lysate environment vs. pure recombinant protein, we would not expect the first to follow exactly the same distribution as either our GS or BS recombinant standard and would rather focus on the overall qualitative trend of eluting in earlier fractions. However, we would like to emphasize that demonstrating the oligomeric state of a protein in the cell/lysate is technically a very demanding task that is rarely attempted. Exploring new ways of analyzing it (e.g., fluorescence cross-correlation spectroscopy), especially in bacteria, is planned by us as a future project but is very challenging. Moreover, even with the most sophisticated techniques, the uncertainty between homo-oligomers and other types of complexes/interactions would remain.

In our view, the strength of our study lies in the in-depth in vitro analysis of the spectrum of possible McsB oligomers, the rules governing their interconversion, and the functional consequences, in which we rely not only on SEC but also on mass photometry (which has a much finer resolution) and in the case of the octamer – on X-ray crystallography. While unable to conclusively show that these states occur in vivo due to technical limitations as described above, we would like to point to the strong phenotype of the R194K mutation, which, due to the conservative character of the mutation and its location away from the active site would be difficult to rationalize in terms other than changes in the oligomeric state.

The proposed negative-stain EM experiment could be a potential route for further investigation, but an exact identification of species based on visual inspection of EM micrographs, given the heterogeneity of bacterial cell lysates, seems similarly ambiguous. We have attempted negative stain analysis of recombinant McsB (*B. subtilis*) but found that even with the pure recombinant protein the multi-facetted oligomeric ensemble of different shapes is very difficult to characterize (Author response image 1). We believe that we would not be able to draw any conclusions from such an experiment performed with endogenous McsB present in or pulled out of the bacterial cell lysate.

**Author response image 1. respfig1:** Negative-stain EM of recombinant McsB enriched in HOOs (higher-order oligomers). Bar; 200 nM. Staining was performed with 2% uranyl acetate. Measured on a FEI Morgagni 268D microscope operated at 80 kV.

In sum, the original purpose of Figure 1c was to see a potential redistribution of McsB species upon heat-shock in cells and offer indirect support for in vivo occurrence of the McsB oligomers. Consistent with the rules of experimentation, we used an experiment that could have disproved/falsified the importance of oligomers by showing that McsB remains largely mono or dimeric in the lysate. Instead, especially under heat shock, we observed the protein in fractions corresponding to high molecular weight species, which leaves open the possibility that the kinase forms high oligomers in the cell, even if the uncertainty remains.

However, we agree that this experiment was less conclusive than we made it seem in the text. Hence, we adjusted our text so that we phrase our interpretation of the Western blot profiles more carefully and present the two equally possible scenarios (i.e. dimer shift caused by homo-oligomerization or dimer shift caused by association with substrates). Therefore, we changed the paragraph describing Figure 1c to further highlight the limitation of our method of choice and the need for further analysis:

“To explore whether these higher-order kinase forms might be present in vivo, we analyzed *B. subtilis* cells grown under normal and heat-shock conditions. After disrupting the bacteria, we applied the cell lysate to a size exclusion chromatography (SEC) column, separating the bacterial proteins according to size. The Western blot profiles of the endogenous protein obtained using an McsB antibody were compared with those of recombinant McsBGS (isolated dimer) and McsBBS (enriched in HOOs), serving as our 80 kDa and 400 kDa size standards, respectively. This analysis indicated that endogenous cellular McsBBS forms assemblies of various sizes (Figure 1c). Intriguingly, McsBBS showed different distributions depending on the investigated environmental scenarios. Under standard conditions, the bulk of McsBBS was present as smaller species with sizes comparable to recombinant dimers, whereas heat-shock induced upregulation of markedly larger complexes. The size shift could indicate oligomerization of McsB into species similar to those found for the in vitro reconstituted McsBBS protein. Equally likely, these larger objects could be McsB dimers associated with folded/unfolded substrates, or with substrates whose abundance is up-regulated during heat-shock.

The presence of McsB in fractions at the SEC column’s void volume might point to the induced formation of larger oligomers of the protein kinase that may be associated with soluble protein aggregates, its preferred substrates in vivo (Trentini et al., 2016). In sum, these data suggest that in the cell, the McsB kinase exists in different complexes, which interconvert in response to changing environmental conditions. We consider it likely that these states include the oligomeric states recapitulated in vitro, but the nature of McsB associations in vivo should be further probed with other techniques in the future.” (Results, page 6)

3. New data on RRAA mutant: In Figure 5c, d it isn't clear what they are using for "HOO" – is it a particular fraction or the entire HOO profile. That is, does this sample still contain dimer? This should be clarified.

We would like to apologize for not being explicit enough when describing Figure 5c, d. The HOO sample is enriched in higher oligomeric species (entire HOO profile), as seen in Figure 5c, but the referee is right, our RRAA sample still contains dimeric McsB. Considering that this partial enrichment in HOOs already leads to significantly elevated activity, the activity of pure HOOs can be expected to be considerably higher than that of the dimer, due to allosteric auto-activation of the kinase upon multimerization (Figure 5d).

We have adopted the manuscript accordingly. It now reads as follows:

“Inherent to the dynamic nature of the open oligomer vs dimer, the HOO fraction also contains dimeric McsB (Figure 5c). Therefore, considering that a sample enriched in HOO particles exhibits significantly elevated activity, the activity of pure HOOs should be markedly higher than that of the dimer (Figure 5d). These data show that the pArg:pR-RS* mediated linkage of McsB subunits exerts a stimulatory effect on kinase activity, which is consistent with the previously described allosteric effect of binding pArg-containing ligands (Suskiewicz et al., 2019).” (Results, page 13)

To be more precise, we also changed the figure legend of Figure 5c,d:

“(c) Secondary SEC runs of the McsBBS R190A/R194A mutant enriched in either HOOs (higher-order oligomers) or dimers at 20 µM concentration”.

“(d) Radiometric kinase assay of wildtype McsBBS and the McsBBS R190A/R194A mutant enriched in either HOOs (higher-order oligomers) or dimers (1 µM) against the folded model protein UNC-45core”.

4. The δ ywlE heat shock experiments on cell viability are convincing. The size distribution could not be assessed per Figure 6Sb, but why weren't these done under heat shock conditions – wouldn't this make the difference more obvious? Also, what are the higher bands ~55kDa in the Figure 6 supplement 2 western blots?

We apologize that by stating “… of heat-shocked wildtype and ΔywlE *B. subtilis* cell lysates …” in the figure legend of Figure 6 —figure supplement 2 we misleadingly gave the impression that the experiment was performed at optimal growth temperature and only the cellular lysate was incubated at 53 degrees Celsius. In reality, it is the cell cultures (both wild-type and δ-ywlE) that were exposed to the heat shock. We thus changed the text to “… lysate of heat-shocked wildtype and ΔywlE *B. subtilis* cultures…” to remedy this misconception. The figure legend of Figure 1c was also changed accordingly.

The antibody used in the manuscript was a polyclonal antibody raised against recombinantly expressed and purified McsB from *B. subtilis* and the 55 kDa, as well as the 35 kDa band, is a non-specific cross-reaction with another cellular protein enriched in some fractions, which we have also observed in δ-mcsB *B. subtilis* lysates, suggesting it does not belong to McsB (Author response image 2).

**Author response image 2. respfig2:** Anti-mcsB Western blot of wildtype and δ-mcsB *B. subtilis* at different heat-shock durations. Red arrows indicate the unspecific 55 and 35 kDa bands.